# Evolutionary Origins of DNA Repair Pathways: Role of Oxygen Catastrophe in the Emergence of DNA Glycosylases

**DOI:** 10.3390/cells10071591

**Published:** 2021-06-24

**Authors:** Paulina Prorok, Inga R. Grin, Bakhyt T. Matkarimov, Alexander A. Ishchenko, Jacques Laval, Dmitry O. Zharkov, Murat Saparbaev

**Affiliations:** 1Department of Biology, Technical University of Darmstadt, 64287 Darmstadt, Germany; paulina.prorok@tu-darmstadt.de; 2SB RAS Institute of Chemical Biology and Fundamental Medicine, 8 Lavrentieva Ave., 630090 Novosibirsk, Russia; grin.inga@gmail.com; 3Center for Advanced Biomedical Research, Department of Natural Sciences, Novosibirsk State University, 2 Pirogova St., 630090 Novosibirsk, Russia; 4National Laboratory Astana, Nazarbayev University, Nur-Sultan 010000, Kazakhstan; bmatkarimov@nu.edu.kz; 5Groupe «Mechanisms of DNA Repair and Carcinogenesis», Equipe Labellisée LIGUE 2016, CNRS UMR9019, Université Paris-Saclay, Gustave Roussy Cancer Campus, F-94805 Villejuif, France; Alexander.ISHCHENKO@gustaveroussy.fr (A.A.I.); laval-j@orange.fr (J.L.)

**Keywords:** DNA repair, DNA glycosylases, AP endonucleases, protein folds, structural homology

## Abstract

It was proposed that the last universal common ancestor (LUCA) evolved under high temperatures in an oxygen-free environment, similar to those found in deep-sea vents and on volcanic slopes. Therefore, spontaneous DNA decay, such as base loss and cytosine deamination, was the major factor affecting LUCA’s genome integrity. Cosmic radiation due to Earth’s weak magnetic field and alkylating metabolic radicals added to these threats. Here, we propose that ancient forms of life had only two distinct repair mechanisms: versatile apurinic/apyrimidinic (AP) endonucleases to cope with both AP sites and deaminated residues, and enzymes catalyzing the direct reversal of UV and alkylation damage. The absence of uracil–DNA *N*-glycosylases in some Archaea, together with the presence of an AP endonuclease, which can cleave uracil-containing DNA, suggests that the AP endonuclease-initiated nucleotide incision repair (NIR) pathway evolved independently from DNA glycosylase-mediated base excision repair. NIR may be a relic that appeared in an early thermophilic ancestor to counteract spontaneous DNA damage. We hypothesize that a rise in the oxygen level in the Earth’s atmosphere ~2 Ga triggered the narrow specialization of AP endonucleases and DNA glycosylases to cope efficiently with a widened array of oxidative base damage and complex DNA lesions.

## 1. Introduction

This theoretical essay was inspired by Miroslav Radman’s works on the mechanisms of SOS response and mismatch repair in bacteria. In their seminal paper, Watson and Crick did not point out the key role of the double-stranded DNA structure for maintaining genome stability thanks to DNA repair. Indeed, to carry out its hereditary function, DNA requires not only replication for self-reproduction and transcription for converting genetic information into RNA and then to proteins, but also repair, to fix eventual DNA damage that otherwise can induce mutations and block template reading. As shown in Figure 1, cellular life evolved several distinct enzymatic mechanisms to counteract mutagenesis and DNA damage to their double-stranded genomes, such as photolyase-catalyzed direct damage reversal (DR) (Figure 1A), the removal of damaged residues via DNA excision (Figure 1B–J), specialized DNA polymerase-catalyzed lesion tolerance/bypass, nucleotide pool sanitization by phosphorylases, and recombinational repair by RecA-like proteins. It should be stressed that the mechanistic principles used in each repair strategy are fundamentally different. In the DR and nucleotide pool sanitization pathways, both of which do not require a double-stranded structure, DNA damage is either removed or prevented without the need for the excision of modified nucleotides and de novo DNA synthesis (Figure 1A). DNA damage tolerance strategy employs both template switching and lesion bypass and, similar to DR, does not require the excision of the damaged base. In contrast, DNA excision repair, which includes pathways such as the base excision repair (BER) (G-H), nucleotide incision repair (NIR) (F), alternative excision repair (AER) (I), nucleotide excision repair (NER) (J), and mismatch repair (MMR), proceeds through phosphodiester bond cleavage, the removal of DNA damage via excision, de novo DNA synthesis with the non-damaged strand as a template, and ligation. Recombination repair also relies on de novo synthesis but the template is provided by another copy of double-stranded DNA.

Previously, several excellent bioinformatics studies discussed the origin of DNA repair. Perhaps the first phylogenomic analysis of DNA repair proteins, based on a limited set of the complete genome sequences, was performed by Eisen and Hanawalt, who made predictions about the repair phenotypes and offered important insights into the evolution of whole pathways [1]. They suggested that photolyases, alkyltransferases, Xth/Nfo AP endonucleases and MutY/Nth DNA glycosylases are very ancient enzymes and were present in the common ancestor, whereas the NER pathway had different origins in bacteria and eukaryotes/archaea. A more recent phylogenomic study by Goosen and Moolenaar confirmed the diverse origins of the NER pathway in bacteria and eukaryotes, and the lack of the dedicated NER machinery in archaea [2]. Several recent review papers on the origin of the photolyase/cryptochrome family describe the evolution of these enzymes to perform diverse functions [3,4]. The convergent evolution of six structural superfamilies of DNA glycosylases is discussed in excellent reviews by Krokan’s and Wallace’s laboratories [5,6,7].

The very multiplicity of DNA repair pathways begs the question of how they, particularly excision repair now in charge of the elimination of deleterious DNA base lesions, could emerge very early in the evolution? Could the copious details known about the molecular mechanisms of repair pathways yield insights into hypothetical DNA repair mechanisms in the last universal common ancestor (LUCA) of all living organisms? Obviously, the genotoxic factors faced by the most ancient life forms depend both on the environment of the early Earth and on the metabolic reactions used by life then. The earliest commonly accepted geochemical evidence of life is dated at least ~3.77 Ga, whereas molecular evidence suggests a date >3.9 Ga, before the Late Heavy Bombardment events [8,9,10]. For a significant part of the early biotic evolution, life was likely exposed to hostile conditions such as high temperatures, extreme pH, an anoxic UV-transparent atmosphere and Earth’s weak magnetic field. The genetic evidence also suggests that the metabolism in LUCA was replete with carbon radical-based reactions [11,12]. Thus, several candidates for primeval genotoxic factors appear feasible: spontaneous DNA hydrolysis, short-wavelength UV (UVC) light, ionizing radiation and alkylating compounds. On the other hand, there is a general agreement that the oxygen-rich atmosphere and widespread oxidative metabolism appeared much later in the history of Earth, thus making oxidative DNA damage of substantially less concern than it is today. Abundant geological evidence agree on a dramatic rise in the atmospheric oxygen approximately ~2.4–2.0 Ga [13,14].

The main products of spontaneous DNA decay in water are apurinic/apyrimidinic (AP) sites, generated by the hydrolysis of *N*-glycosidic bonds, mainly in purine deoxynucleotides, and uracil (U) residues, arising from the hydrolytic deamination of cytosines; hypoxanthine (Hx) and xanthine (Xan) residues are also produced by the deamination of adenine and guanine, respectively, but with lower yields (Figure 2A). The main products of UV radiation are cyclobutane pyrimidine dimers (CPDs) and pyrimidine–pyrimidone (6–4) photoproducts (6–4PPs) (Figure 2B) [15]. Both photoproducts are cytotoxic (block DNA replication and transcription) and mutagenic. Notably, fossilized intertidal microbial mats with evidence of periodic atmosphere exposure suggest that the pathways to cope with UV damage were already in place as early as ~3.4 Ga [16]. Later in the history of Earth, the production of oxygen during photosynthesis resulted in the formation of the ozone layer, which efficiently shielded life from lethal UV irradiation. DNA strand breaks with 3′-blocking groups, abasic sites and modified bases are major products of ionizing radiation [17,18]. Pyrimidine hydrates in DNA such as 6-hydroxy-5,6-dihydrothymine occur spontaneously and can also be generated by exposure to UV light and ionizing radiation under anoxic conditions (Figure 2A) [19,20,21]. Electrophilic agents, including carbon radicals, avidly react with electron-rich positions of nucleobases, forming several classes of their *N*- and *O*-alkylated derivatives [22,23]. Last but not least, the replication machinery is not a perfect copier and can make mistakes, misincorporating ribonucleotides or non-complementary dNMPs [24,25].

At least four major DNA repair pathways can efficiently deal with these abnormal chemical structures in DNA: photolyase-, alkyltransferase- and dioxygenase-catalyzed DR, DNA glycosylase-initiated BER, AP endonuclease-mediated NIR and multiprotein complex-dependent NER (Figure 1). The DR of a damaged base to its native state without excision and de novo DNA synthesis provides the simplest, most efficient and accurate biochemical mechanism of DNA repair. However, DR mechanisms are known only for a handful of UV-induced and alkylated base lesions. NER is the main pathway used by present-day cells to remove a variety of structurally unrelated bulky DNA lesions such as those generated by UV radiation and xenobiotics including environmental toxins and anti-cancer drugs. Here, DNA lesions are eliminated by a multiprotein complex through dual incision bracketing the adduct in the damaged strand to form a short oligonucleotide containing the lesion and a single-stranded gap serving as a template to restore DNA integrity through the action of DNA polymerases and ligases.

The classic BER pathway is initiated by a DNA glycosylase excising a modified or mismatched base by the hydrolysis of its *N*-glycosidic bond, generating a free base and an AP site, which is a cytotoxic and mutagenic repair intermediate (Figure 1G). Then, either a hydrolytic AP endonuclease or AP lyase activity present in certain DNA glycosylases nicks DNA 5′ or 3′ to the AP site, respectively (Figure 1B,C). The incised DNA is further processed by either 2′-deoxyribo-5′-phosphate lyase (dRpase) or 3′-cleaning repair activity (3′-phosphodiesterase or 3′-phosphatase), and then by a DNA polymerase, DNA ligase, and other accessory proteins to restore the integrity of DNA. The flow of the DNA glycosylase-initiated BER, which requires the sequential action of two enzymes for proper DNA incision, must be tightly coordinated because of the genotoxic intermediates generated. In the alternative DNA glycosylase-independent NIR pathway, an AP endonuclease directly cleaves damaged DNA 5′ to various modified nucleotides, generating 3′-OH and 5′-phosphate termini, thus avoiding potentially genotoxic AP site formation (Figure 1F) [26]. The NIR pathway, which can be genetically separated from BER, is evolutionarily conserved in all three domains of life [27,28]. BER and NIR are the most versatile pathways that recognize a wide range of DNA modifications from small, non-bulky ones to bulky lesions that distort the helical DNA structure.

From the mechanistic point of view, the DR and NIR pathways are the simplest, requiring few steps and less energy as compared to the BER and NER machineries, which employ multiple sequential steps and more than one protein to cleave damaged DNA and generate proper ends. Recent findings shed light on the evolutionary origins of DNA repair pathways and mechanisms of DNA damage recognition. In this scenario, following the Oxygen Catastrophe there was an increase in the spectrum and complexity of DNA damage, which overwhelmed the capacity of the ancient DR and NIR system to maintain genome stability and resulted in the appearance of new pathways such as BER and NER. Here, we propose that the DNA glycosylase-initiated BER and multi-protein complex mediated NER systems appeared later in evolution, offering more versatile and efficient catalytic mechanisms that involve sequential actions of several repair proteins to remove DNA damage and generate a single-stranded gap of varying length with a proper 3′-end that can be used as a primer for DNA repair synthesis and ligation (Figure 1G,H,J). Here, we advance the idea that under the conditions on early Earth, the spectrum of DNA damage was limited and only few singular repair pathways such as DR and NIR were sufficient to protect the genome of LUCA. We propose that LUCA employed a photolyase and *O*^6^-alkylguanine alkyl transferase to repair UV and alkylation DNA damage, respectively, whereas spontaneous DNA decay and ionizing radiation-induced DNA damage were counteracted by the NIR-specific AP endonucleases, AP lyases and 3′-repair phosphodiesterases (Figure 1B–F).

## 2. Types of DNA Damage

Cellular macromolecules such as proteins, lipids and nucleic acids are prone to spontaneous decomposition because of their intrinsic chemical instability. In addition, endogenous oxidative stress and a number of exogenous environmental factors, such as alkylating and oxidizing agents, photosensitizers, antibiotics, and ionizing and UV radiation, also contribute to structural damage to cellular components. Cellular DNA constantly undergoes structural alterations, resulting in DNA lesions that are cytotoxic, miscoding, or both and are believed to be at the origin of cell death and mutations [29,30]. DNA damage can be classified by its nature: spontaneous vs. induced, and by the structure of the produced lesions: singular vs. complex.

### 2.1. Hydrolytic DNA Damage

As today, ancestral life would encounter spontaneous DNA decay as a primary challenge to genetic inheritance, further promoted by elevated temperatures. In aqueous solutions, sufficiently long DNA molecules are prone to spontaneous *N*-glycosidic bond hydrolysis and deamination at significant rates [31,32] (Figure 2A). Under normal physiological conditions, ~1 per 10^5^ purines are lost from DNA per day, generating AP sites [31]. AP sites have miscoding properties, since the replication machinery follows the “A rule”, preferentially incorporating dAMP opposite them [33,34]. Spontaneous loss of the exocyclic amino groups in C, A and G bases generates highly mutagenic U, Hx and Xan residues, which induce C → T, A → G, and G → A transitions, respectively (Figure 2A) [35,36,37]. Under typical intracellular conditions, deamination in DNA occurs in about 1 per 10^7^ C residues per day, whereas purines are deaminated at ~1/10 of this rate [32]. Importantly, C deamination in single-stranded nucleic acids is an order of magnitude faster than in duplex DNA, and the need to distinguish an instructive genomic base from the damage product is believed to play a critical role in T substituting for U upon transition from the RNA world to the modern DNA world [38].

### 2.2. UV Damage

Solar light is, and was, the primary source of UV radiation from the very beginning of life on Earth. The UV radiation spectrum can be divided into three bands: the least energetic long-wavelength UVA (315–400 nm), medium-wavelength UVB (280–315 nm), and the most energetic short-wave UVC (100–280 nm). UV radiation, especially UVC, is very efficient at killing bacteria, and germicidal UVC-emitting lamps are frequently used for sterilization purposes. The most frequent and detrimental effect in DNA of UVC and UVB is the formation of pyrimidine dimers (Figure 2B), first discovered over sixty years ago by Beukers and Berends [39]. The two most common UV products are CPDs formed by the cyclization of C5=C6 bonds of two adjacent pyrimidines, and 6–4PPs in which two adjacent pyrimidines are connected by a covalent bond between C6 and C4 (Figure 2B). 6–4PPs can be converted into Dewar valence isomers upon UVB irradiation [40]. CPDs are more frequent than 6–4PPs and constitute ~80% of the total UV-induced lesions [15]. In addition to bulky DNA adducts, UV radiation induces genotoxic small base modifications—pyrimidine hydrates such as 6-hydroxy-5,6-dihydrothymine, 6-hydroxy-5,6-dihydrocytosine and 6-hydroxy-5,6-dihydrouracil (Figure 2A) [41,42]. The appearance of oxygen in the atmosphere ~2.4 Ga due to the photosynthetic activity of the first cyanobacteria led to the formation of the ozone layer, which absorbs radiation below ~310 nm, thus blocking all UVC and 95% of UVB from the surface of the Earth and offering efficient protection for living organisms from excessive UV radiation [13,14].

### 2.3. Ionizing Radiation Damage

Ionizing radiation consists of subatomic particles or electromagnetic waves that are sufficiently energetic to tear electrons off atoms or molecules. Cosmic rays and the decay of radioactive isotopes are the primary sources of natural ionizing radiation on Earth, contributing to background radiation. Since the Earth’s core solidified only ~1.5 Ga, the early Earth’s magnetic field was much weaker than it is today, resulting in ~100-fold higher solar high-energy flux [43,44]. Ionizing radiation induces high-energy radiolysis of water molecules, producing highly reactive species such as hydrated electrons, ^•^H and ^•^OH radicals. The major products generated by ionizing radiation are DNA strand breaks with 3′-phosphoglycolate and 3′-phosphate termini, which arise by the abstraction of a hydrogen from deoxyribose at C1′ or C4′ by ^•^OH radicals following base loss and sugar fragmentation [45]. A salient feature of ionizing radiation is the generation of closely spaced DNA lesions forming clusters, i.e., two or more damaged bases or strand breaks spaced by less than one helical turn of duplex DNA, either on the same strand or on the opposite DNA strands [17]. It should be stressed that exposure to IR under low oxygen concentration favors the formation of 5,6-dihydrothymine (DHT), thymine hydrates, α-anomeric 2′-deoxynucleotides (αdN) and 8,5′-cyclopurine deoxynucleosides, but not the classic oxidized bases such as 7,8-dihydro-8-oxoguanine (8oxoG), 5-hydroxycytosine (OHC) or thymine glycol (Tg) (Figure 2A) [46,47,48,49,50,51,52]. This might suggest that such “anoxic” lesions were more physiologically relevant as damage generated by ionizing radiation in cellular DNA on the early Earth.

### 2.4. Alkylation Damage

Nitrosamines are formed under acidic pH by a reaction between nitrates or nitrites and secondary amines generated during protein degradation. Most nitrosamines are potent mutagens and carcinogens. The metabolic activation of nitrosamines converts them to alkylating electrophilic compounds that react with DNA bases at all exocyclic oxygens and most ring nitrogens [53,54]. These agents can also alkylate non-bridging oxygen atoms of internucleoside phosphates, generating alkylphosphotriesters. S_N_1 alkylating agents, such as *N*-methyl-*N*-nitrosourea (MNU) and *N*-methyl-*N’*-nitro-*N*-nitrosoguanidine (MNNG), use a monomolecular mechanism to alkylate at N and O atoms, while the S_N_2 agents, which include methyl methanesulfonate (MMS) and dimethylsulfate (DMS), act through a bimolecular mechanism to alkylate at N atoms only. Ethyl methanesulfonate (EMS) uses both S_N_1 and S_N_2 mechanisms. 7-methylguanine (7meG), 3-methyladenine (3meA) and *O*^6^-methylguanine (*O*^6^meG) are the most abundant alkylated lesions [55,56]. 1-Methyladenine (1meA) and 3-methylcytosine (3meC) are also generated by S_N_2 agents, such as MMS, preferentially in single-stranded DNA and RNA [57]. The presence of 3meA, 1meA and 3meC residues in the DNA template can block DNA replication, making them highly cytotoxic lesions, whereas 7meG is non-toxic and non-mutagenic [58,59]. S_N_1 type agents such as MNNG and MNU are the most potent chemical carcinogens because they react more readily with exocyclic oxygen of DNA bases to generate the highly miscoding *O*^6^meG and *O*^4^-methylthymine (*O*^4^meT) adducts [60]. The major adduct *O*^6^meG mispairs with T during DNA replication, resulting in G → A transitions. Interestingly, mutagenic S_N_1 nitroso-compounds can arise endogenously in bacteria through the nitrosation of amino acids, peptides or polyamines [61]. Endogenous methyl donor *S*-adenosylmethionine (SAM) can also act as a weak S_N_2 type of alkylating agent in vivo and contribute to endogenous DNA alkylation [62]. Moreover, SAM is cleaved by a variety of enzymes (radical SAM enzymes), producing a 5′-deoxyadenosyl radical that is further used to generate other free radical metabolic species [63]. Radical SAM reactions likely constituted a significant part of LUCA metabolism [12]. Thus, exogenously and endogenously generated alkylating species would present a danger for the genetic material of ancestral life forms.

### 2.5. Oxidative Damage

The appearance of an oxygen-rich atmosphere resulted in the wide adoption of oxidative phosphorylation as the major pathway to produce adenosine triphosphate (ATP). In a canonical electron transport chain, free radicals are generated when an oxygen molecule promiscuously reacts with one of the transported electrons before it reaches the final complex IV (cytochrome c oxidase). The superoxide anion radical (^•^O_2_^−^) is generated, which is then converted to hydrogen peroxide (H_2_O_2_). Reduced transition metals such as Fe^2+^ and Cu^+^ react with H_2_O_2_ to produce a highly reactive hydroxyl radical, ^•^OH, via Fenton reactions [64], which in turn damages cellular DNA, proteins and membrane lipids [65]. About 80 different types of base and sugar lesions induced by these reactive oxygen species (ROS) have been identified [66]. ROS can damage both nucleobases and sugars in DNA either directly or indirectly. Hydroxyl radicals, which are the most active species, predominantly react at C8 of purines forming 8oxoG [67] and imidazole ring-opened products such as 2,6-diamino-5-formamidopyrimidine (FapyG) or 4,6-diamino-5-formamidopyrimidine (FapyA) [68] at the C5=C6 double bond of pyrimidines to form glycols [46,69], and at C8 and C5′ of purine deoxynucleosides forming their 8,5′-cyclopurine derivatives [70]. Major oxidized bases due to endogenous ROS, such as 8oxoG, 5OHC, and 5-hydroxyuracil (5OHU), are miscoding and, if not repaired, lead to mutation upon replication [25,71,72]. Others, such as oxidized AP sites, thymine glycols and fragmentation products of DHT, cause replication block and therefore have a strong cytotoxic effect [73,74,75,76].

Indirectly, ROS can damage DNA through membrane lipid peroxidation (LPO), generating highly reactive α,β-unsaturated aldehydes (enals), e.g., acrolein, crotonaldehyde, and 4-hydroxynonenal (4-HNE), which can react with nucleobases, forming exocyclic adducts such as 1,*N*^2^-γ-hydroxypropano-dG [77,78,79]. When present in DNA, these exocyclic adducts can further yield inter-strand DNA crosslinks (ICLs), often between guanines in the CpG context, as a result of exocyclic ring opening and interaction with undamaged guanine in the opposite DNA strand (Figure 2B) [80]. ICLs are highly lethal DNA lesions that block DNA replication, transcription and recombination by preventing strand separation. In addition to ICLs, chemical agents such as formaldehyde and physical factors such as ionizing and UV radiation can crosslink proteins to an undisrupted DNA strand, generating DNA–protein crosslinks (DPC) [81].

## 3. DNA Repair in Reconstructed and Minimal Genomes

The explosive growth of the number of fully or partially sequenced genomes and the appearance of tools to establish their evolutionary relationships led to several attempts to reconstruct both minimal genomes allowing modern cells to survive [82,83,84,85,86,87,88] and putative ancestral genomes [12,89,90,91]. The last universal common ancestor is a hypothetical organism or, more precisely, a common gene pool from which all presently living organisms descend [92,93,94]. A recent reconstruction of the LUCA genome outlined 355 protein families potentially present in this pool [12]. Table 1 lists representative proteins from *E. coli* and humans that are related to DNA repair in today’s species and have their putative ancestry among the LUCA genes.

Several notes of caution are imperative when trying to use these reconstructions to understand the evolution of DNA repair systems. Minimal genomes are based on experimental data from a very limited number of bacterial species and thus necessarily reflect the kinds of genotoxic challenges faced by them under laboratory growth conditions. Even if the gene inactivation experiments would be performed in another environment, or with other species, the gene sets identified as essential might be different. Reconstructed ancestral genomes, on the other hand, strongly depend on assumed evolutionary trees of the major phylogenetic groups included in the analysis, which are still far from being firmly established. The early evolutionary split between Bacteria and Archaea is especially obscure, hindering the analysis of gene ancestry and lateral transfer in the eukaryotic lineage [95,96,97]. For example, attempts to uncover eukaryotic genes of bacterial origin produced widely divergent results, at least concerning DNA repair genes [98,99]. It is also important to realize that the reconstructed LUCA genome reflects the metabolism and environment characteristic of the last genetic bottleneck, which very likely far post-dated both the appearance of life and its transition to the DNA world.

That said, and despite little congruence between essential DNA repair genes in minimal and reconstructed ancestral genomes evident from Table 1, these exercises may be useful to understand the biology of LUCA and how it coped with genome damage. A recent reconstruction focused on LUCA’s metabolism revealed an abundance of iron–sulfur (FeS) clusters, radical SAM reactions, and a variety of nucleobase modifications [12]. FeS clusters are important modules in many DNA repair enzymes from BER and NER pathways and in some DNA polymerases, often serving as at-a-distance sensors of oxidative DNA damage [100,101]. Radical SAM enzymes catalyze a diverse set of reactions and are based on the ancient TIM barrel fold, which also includes one major group of AP endonucleases [63,102,103]. Among other cofactors, LUCA appears to rely on flavin and pterin moieties, critical elements in UV-protecting DR. Finally, LUCA’s genetic makeup is consistent with the hot environment, making hydrolytic reactions—base loss and deamination—possibly the strongest threat for its genome.

## 4. Spontaneous DNA Damage: To BER or Not to BER?

### 4.1. DNA Glycosylases: Primordial Players or a Later Adaptation? 

Today, deaminated bases (mostly U) are removed by one of multiple uracil–DNA glycosylases, whereas AP sites, either spontaneous or glycosylase-formed, are nicked by an AP endonuclease (Figure 1B,G). However, the appearance of toxic and mutagenic AP sites in the course of canonical BER raises the question of why such a potentially hazardous pathway evolved at all. A nuclease activity recognizing a lesion and cleaving DNA next to it (Figure 1E,F and see next sections) would not be entailed with this drawback. However, uracil–DNA glycosylase is one of few enzymes that both apparently existed in LUCA and are essential in minimal genomes (Table 1). Did ancestral DNA repair pathways require DNA glycosylases?

Notably, at least seven protein folds can catalyze base excision from DNA: the uracil–DNA glycosylase-like α/β-fold, helix–hairpin–helix motif (HhH)-containing enzymes, helix–two-turn–helix motif (H2tH)-containing enzymes, the formyltransferase C-terminal domain fold, the T4 pyrimidine dimer DNA glycosylase fold, the tetratricopeptide-like repeats fold, and the HALFPIPE fold [104,105]. Even more strikingly, none of the known DNA glycosylases are based on structural folds of ubiquitous metabolic enzymes capable of hydrolyzing *N*-glycosidic bonds between sugars and aromatic heterocycles (including nucleobases), such as nucleoside hydrolases, purine/uridine nucleoside phosphorylases, or *N*-(deoxy)ribosyltransferases, or the more specialized SMF/DprA-LOG superfamily that includes enzymes producing heterocyclic signaling molecules, or ribosome-inactivating RNA *N*-glycohydrolases. One is tempted to speculate that the weaker nature of the *N*-glycosidic bond in DNA compared with that in RNA makes its hydrolysis so inherently easy that many DNA-binding proteins could evolve an ability to catalyze this reaction with biologically acceptable rates. It is interesting to note that almost all DNA glycosylases and several other structurally unrelated classes of proteins (e.g., AP endonucleases, nucleobase dioxygenases, DNA methyltransferases, and epigenetic readers) kink DNA and flip out the target nucleotide [106,107], and it has been suggested based on molecular dynamics modeling that the configuration of DNA phosphates in this strained intermediate mainly provides the catalytic power for the *N*-glycosidic bond breakage [108]. An intriguing corollary of the differential *N*-glycosidic bond stability was noted by Rios and Tor [109], who hypothesized that the RNA world used an extended repertoire of nucleobases due to the need to combine information storage and catalysis in one type of nucleic acid, and the primeval glycosylases might have evolved with the appearance of DNA genomes to eliminate non-standard bases no longer needed, including U, from the genome carrier. This suggestion aligns both with the large repertoire of nucleobase modifications in LUCA [12] and with the ubiquity of UNG-like enzymes, which seem to represent an ancient uracil-binding fold [110,111]. Interestingly, the widest phylogenetic diversity today can be found not among Family 1 uracil–DNA glycosylases, where extensively characterized human and *E. coli* enzymes belong, but among Family 4 and Family 5 enzymes that were initially discovered in extremophilic archaea and bacteria and contain an FeS cluster [112,113,114,115].

In total, nine families of UNG-like enzymes have been defined so far, of which seven are biochemically characterized (Figure 3). Some of them, such as Family 1, where extensively studied human and *E. coli* enzymes belong, and Family 4, initially discovered in extremophilic archaea and bacteria and containing an FeS cluster, seem to be bona fide uracil–DNA glycosylases with little activity on other substrates. Others, such as SMUG1 (Family 3), FeS-containing Family 5, and the recently discovered SMUG2 and *Bradyrhizobium diazoefficiens* uracil–DNA glycosylase (BdiUng)-like enzymes, have wider substrate specificity that may additionally include other U derivatives (5OHU, 5-hydroxymethyluracil, 5-formyluracil), Hx, and Xan [116,117,118,119]. Thymine–DNA glycosylase (TDG) present in animals and fungi is involved in active epigenetic demethylation, removing oxidized and/or deaminated derivatives of 5-methylcytosine [120,121], and together with its bacterial homolog Mug may be the primary glycosylase for exocyclic pyrimidine adducts such as 3,*N*^4^-methylcytosine, or for 7,8-dihydro-8-oxoadenine (8oxoA) [122,123]. Finally, Family 6 lacks uracil-removing activity completely, excising hypoxanthine instead [114].

The most versatile group of DNA glycosylases is the HhH superfamily, the members of which are present in all domains of life, remove all major kinds of damaged bases and some mismatched bases, and can be monofunctional or have an associated β-elimination activity (Figure 1H). The HhH motif itself is a small non-specific DNA-binding element also found in DNA and RNA polymerases, transposases, Holliday junction branch migration protein RuvA, RecD helicase and several other DNA-binding proteins [126,127,128]. In DNA glycosylases with an associated β-elimination activity (Nth, OGG1, OGG2), the second helix bears a catalytic Lys residue that acts as a nucleophile to displace the target base. The homologous Lys in DNA polymerases β and λ serves as a nucleophile in the dRpase reaction catalyzed by these enzymes [129,130]. In monofunctional enzymes (MutY, AlkA, MBD4), this Lys is replaced with different non-basic residues, indicating that the ability to perform β-elimination was likely ancestral in the glycosylases belonging to this superfamily. In DNA glycosylases, the HhH motif is followed by a Gly/Pro-rich stretch bearing a second catalytic residue, an Asp side chain (GPD motif) [131,132], absent from the non-glycosylase members of the superfamily. Some HhH DNA glycosylases also contain an FeS cluster. Thus, HhH DNA glycosylases likely arose as a result of the evolution of an ancient universal DNA-binding HhH motif.

The last major group of DNA glycosylases is built on another small DNA-binding module, the H2tH motif. Unlike the HhH motif, it is found only in a limited group of polypeptides, including ribosomal protein S13 (S18 in eukaryotes) and several ribosome quality control proteins. These glycosylases also carry a DNA-binding β-ribbon zinc finger or a structurally equivalent β_2_-module called a “zincless finger”, and a unique N-terminal β-sandwich domain with the catalytic *N*-terminal amino group and a catalytic acidic side chain [133,134]. Members of this group are missing from Archaea, and, as they are exclusively involved in oxidative damage repair, the H2tH glycosylases represent good candidates for acquisition from symbiotic mitochondria functionally associated with the evolution of oxidative metabolism.

### 4.2. AP Endonucleases: The Archetypal Repair Endonucleases

The occurrence of an enzyme recognizing AP sites in *E. coli* was first described in the seminal work of Verly [135]. AP endonucleases cleave DNA 5′ to an AP site by a hydrolytic mechanism [136,137]. The majority of known hydrolytic AP endonucleases are divided into two distinct families based on amino acid sequence identity to either *E. coli* exonuclease III (Xth) or endonuclease IV (Nfo) [138]. Mg^2+^-dependent Xth-like AP endonucleases belong to a large endonuclease/exonuclease/phosphatase (EEP) superfamily that also includes DNase I, tyrosyl–DNA phosphodiesterase 2, retrotransposon endonucleases and various phosphoester hydrolysis enzymes [139,140,141,142,143]. Nfo-like AP endonucleases are totally different in their structure, belonging to the multifunctional TIM barrel superfamily, which probably originally contained sugar metabolism enzymes [102,144], and utilize Zn^2+^ ions for catalysis. Despite these differences, the geometry of the active sites in Xth- and Nfo-like AP endonucleases is very similar, underlying a convergent catalytic mechanism [145].

Genetic data indicate that AP endonuclease-deficient bacterial, yeast and mammalian mutant cells are highly sensitive to alkylating agents, oxidative stress and ionizing radiation, indicating the critical role of these enzymes in protecting cells from AP sites and DNA strand breaks with 3′-blocking groups [146,147]. Furthermore, the complete lack of AP site repair enzymes in *S. cerevisiae* is lethal, indicating that the background level of AP sites is sufficient to cause cell death, at least in yeast [148,149]. Interestingly, an Nfo-like AP endonuclease is present in a minimal *Mycoplasma* genome and in one of the LUCA genome reconstructions (Table 1). Taken together, these observations suggest that LUCA contained an AP endonuclease (possibly Nfo-like) to counteract the effects of spontaneous base loss in its genome. However, as we argue below, AP sites might not be its only, and even not the primary substrate.

### 4.3. Archaeal AP Endonuclease Mth212 in the Repair of Deaminated DNA Bases

C → T transitions, likely arising through cytosine deamination, are the most frequently occurring base substitutions observed in living organisms [150,151,152]. In addition, uracil in DNA can be generated through the incorporation of dUMP instead of dTMP from the nucleotide pool; this process results in U:A pairs that are not mutagenic but can be lethal when accumulated at a high level in genomic DNA [153,154]. Indeed, as the dU(M/D/T)P pool is a metabolic precursor of thymine nucleotides, the threat of dUMP incorporation is unavoidable, and the need to counteract the buildup of genomic U may be the main reason for the universal persistence of UNG across all major domains of life, including some viruses.

Surprisingly, no homologues of the major UNG-like families could be detected in the complete genome sequences coming from several orders of Euryarchaeota such as Methanopyrales [155], Methanococcales (*Methanocaldococcus jannaschii* [156], *Methanococcus maripaludis* [157]), and two species of Methanobacteriales (*Methanothermobacter thermautotrophicus* [158] and *Methanosphaera stadtmanae* [159]), suggesting the existence of alternative mechanisms for the removal of deaminated DNA bases. The analysis of protein extracts from *M. thermautotrophicus* revealed a nicking activity towards U in ssDNA and dsDNA [27]. The purified protein cleaved the 5′-phosphodiester bond next to dU in a Mg^2+^-dependent manner, and was identified as Mth212, a distinct member of the EEP family. Mth212 cleaves duplex DNA containing an AP site or dU paired with any base, but not DNA with mismatched natural bases. Additionally, the U-endonuclease activity was followed by 3′ → 5′ exonuclease in the lesion vicinity, which extended the nick to a single-stranded gap upstream of the lesion [160]. The repair of dU in dsDNA was successfully reconstituted in vitro with purified enzymes. In the first step, Mth212 cleaves the duplex 5’ next to the lesion and generates the free 3′-OH end extended by DNA polymerase B; the displaced DNA strand is cleaved by 5′-flap endonuclease producing a nick, which is then sealed by DNA ligase to accomplish the repair [161].

The crystal structure of Mth212 exhibits no apparent homology to UNG-like DNA glycosylases but is very similar to Xth-like AP endonucleases. An attempt to obtain the co-crystal structure of Mth212 and dU-containing DNA yielded several abortive enzyme/DNA complexes, in which either one duplex oligonucleotide was bound at both of its ends by two wild-type Mth212 molecules, or two duplex oligonucleotides were bound to one molecule of the catalytically inactive Mth212 D151N mutant, forming a pseudo-continuous DNA duplex [162]. In the latter structure, one oligonucleotide was bound in the same way as with wild-type Mth212, and the second oligonucleotide extended the first one at the enzyme’s active site, resulting in a joint DNA molecule (Figure 4A). Nevertheless, these aberrant Mth212/DNA complexes revealed the mechanism of dU recognition in duplex DNA, surprisingly similar to DNA glycosylases. Mth212 inserts Arg209 amino acid into the DNA base stack and stabilizes the extra-helical flipped-out conformation of dU in a specific active site pocket. The specific recognition of the U base is attained by interactions with Lys125, Ser171 and Gln153. The modeling of dU and dC in the active site of Mth212 showed that these amino acids can discriminate between a damaged and regular DNA base. Thus, nucleotide flipping is a common mechanism used by NIR-specific Xth family AP endonucleases and uracil–DNA glycosylases.

Further characterization of the Mth212 substrate specificity demonstrated that the archaeal repair enzyme, similar to NIR-proficient AP endonucleases such as *E. coli* Nfo and human APE1, is able to cleave at other lesions such as αdA and 5OHC in duplex DNA. Furthermore, under low Mg^2+^ concentrations (≤ 1 mM), APE1 can also cleave duplex DNA 5′ to U opposite G and then extend the resulting nick to a single-stranded gap by its non-specific 3′ → 5′-exonuclease activity [163]. Thus, the direct AP endonuclease-catalyzed DNA incision 5′ to a damaged nucleotide (the NIR pathway) is evolutionarily conserved in Xth-like AP endonucleases from Archaea to humans, and can serve as a back-up—or sometimes the only—mechanism to repair genomic dU residues.

### 4.4. Archaeal Endonuclease Q in the Repair of Deaminated DNA Bases

Another study by Ishino and colleagues showed that cell extracts of hyperthermophilic archaeon *Pyrococcus furiosus* contain a new DNA damage-specific endonuclease, endonuclease Q (EndoQ), which cleaves the phosphodiester bond in DNA at the 5′ side of Hx, U, Xan and AP sites [164]. The repertoire of *P. furiosus* EndoQ (PfuEndoQ) substrates was later extended to include oxidized pyrimidines such as 5,6-dihydrouracil, 5-hydroxyuracil and 5-hydroxycytosine residues, reminiscent of *E. coli* Nfo and human APE1 [165]. The activity of EndoQ is affected by the nature of the base opposite to the lesion [165].

Enzymes belonging to the EndoQ family, not related to other endonucleases, have been identified mainly in the species belonging to Euryarchaeota, such as the extremely hyperthermophilic *Thermococcus kodakarensis*, growing optimally at temperatures over 100 °C, and a mesophilic methanogene *Methanosarcina acetivorans*, which possesses three enzymes for the removal of U from DNA: EndoQ, Xth and Family 4 UNG [166].

Interestingly, it has been shown that EndoQ interacts with PCNA through its PIP-motif, which stimulates the endonucleolytic activity of the repair enzyme [170]. Direct interaction with PCNA as well as activity towards duplex and ssDNA suggest that EndoQ can be a part of both DNA repair machinery and the DNA replication complex. More extensive phylogenic studies permitted the identification of members of the EndoQ family in some bacteria (*Bacillus pumilus* and *B. subtilis),* where the enzymes may participate in antiviral defense mechanisms beside their role in DNA repair [171]. The activity of the *B. subtilis* EndoQ is not inhibited by Ugi, the specific UNG inhibitor from PBS1 phage [165].

Recent X-ray crystallography studies provided insights into the molecular mechanism of EndoQ-catalyzed endonucleolytic cleavage, showing that the damaged base is flipped out into an extrahelical position and placed in the catalytic pocket of the enzyme adjacent to a Zn^2+^-binding site [167]. The enzyme binds three Zn^2+^ ions; two of them reside in the PHP domain and are directly involved in phosphodiester bond hydrolysis, while the third Zn^2+^ is found in a separate Zn-binding domain [167]. More recently, the crystal structures of PfuEndoQ bound to dU-, dI-, and AP site-containing DNA duplexes were solved (Figure 4B) [172]. The highest-resolution structure of PfuEndoQ with the U:G substrate revealed an active site with a single Zn^2+^ ion coordinated by Glu76, His84, His139, and the oxygen atom of the scissile phosphodiester bond [172]. The E76A, H84A, and H139A single amino acid substitutions completely abolish the endonucleolytic activity of EndoQ, suggesting that this Zn^2+^ is essential for phosphodiester bond cleavage [167]. EndoQ binding induces a highly distorted conformation of the damaged DNA strand up- and downstream of the lesion. Three structural domains of EndoQ form a positively charged cleft that accommodates the sharply bent DNA duplex. The damaged nucleotide is at the apex of the kink, rotated out of the duplex and inserted into a deep active site pocket formed by Zn-binding and catalytic domains of the protein (Figure 4B). Interestingly, U and Hx bases are specifically recognized by a different set of amino acid residues, and this mode of recognition is distinct from that of DNA glycosylases. Noteworthy, the His139, Gly169 and Ser171 residues in the catalytic pocket of EndoQ present a steric obstacle for the binding of regular T and A bases because of the presence of a C^5^-methyl group and an N^6^-amino group absent from U and Hx, respectively.

Noteworthy, the nucleotide flipping accompanied with specific interactions in the active site pocket is universally employed by EEP superfamily AP endonucleases, TIM superfamily AP endonucleases, and archaeal EndoQ. Unlike DNA glycosylases and sequence-specific DNA methyltransferases, which have to flip out the target nucleotide in order to get access to the reacting atoms, endonucleases, in principle, could reach the scissile bond without nucleotide eversion. Indeed, structures of substrate and product complexes of DNase I, a non-specific endonuclease also belonging to the EEP superfamily, show no signs of nucleotide eversion [173,174]. Furthermore, closer inspection of the available Nfo/DNA and APE1/DNA structures revealed that these NIR AP endonucleases also utilize the nucleotide flipping mechanism to recognize both AP sites and 3′-terminal nucleotide residues at the DNA strand break (Figure 4C–E). Structural and mutational analysis of EEP superfamily NIR-competent AP endonucleases revealed that amino acid residues in their active site pocket do not interact with damaged DNA bases and instead recognize DNA damage via changes in the sugar puckering and in the distance between two neighboring phosphates in the DNA backbone [175].

Moreover, some residues located outside of the active site (e.g., Thr268 of APE1) can be involved in additional DNA contacts, resulting in a dramatic effect on the NIR activity and Mg^2+^ dependence [168]. Notably, NIR-deficient *E. coli* Xth from the EEP superfamily is a very efficient AP endonuclease and has a more compact active site in comparison to NIR-proficient homologs (APE1, Mth212 and *B. subtilis* ExoA) [175]. Taken together, these observations suggest that nucleotide flipping is a common mechanism of lesion recognition employed by both DNA glycosylases and AP endonucleases. This may be a perfect case when we can trace the convergent evolution of various protein folds to exploit a common physicochemical feature—the inherent thermodynamic instability of damaged DNA.

The idea that repair proteins initially recognize the lesion via the disruption of weakened base stacking and the intercalation of aromatic amino acid side chains into the DNA duplex was proposed for the NER proteins [176,177]. Later, it was demonstrated that the yeast XPC orthologue Rad4 bound to DNA containing a CPD adduct inserts a β-hairpin through the DNA duplex, which enables the two damaged base pairs to flip out of the double helix [178]. Noteworthy, similarly to UVDE, the everted non-damaged nucleotides opposite to CPD are specifically recognized by Rad4, whereas the nucleotides in the dimer become disordered. Furthermore, the crystal structure of the DDB1–DDB2 complex bound to DNA containing a 6-4PP adduct shows that a DDB2 hairpin inserts into the minor groove and extrudes the photodimer into a binding pocket [179]. This specific recognition mechanism enables DDB2 to capture lesions refractory to detection by other damage surveillance proteins. These data suggest that the lesions recognized by the NER proteins are characterized by thermodynamic instability, which in turn facilitates their flipping out by the multi-protein complex machinery.

## 5. Nucleotide Incision Repair: An Ancient and Versatile Mechanism to Counteract Spontaneous DNA Decay and Ionizing Radiation-Induced DNA Damage

### 5.1. NIR as a Remnant of the Pre-BER World

Generally, it is agreed that the environment of early Earth around the time of LUCA’s existence was characterized by high temperatures and a reducing anoxic atmosphere. Under these harsh conditions, the spectra of DNA damage consisted of mainly the products of base loss and deamination, UV light-, ionizing radiation- and alkylation agent-induced DNA lesions with no or very little oxidative DNA damage. The efficiency of archaeal Mth212 and EndoQ-like endonucleases in the repair of deaminated nucleobases without the involvement of DNA glycosylases may suggest that in early evolution, to counteract the deleterious effects of spontaneous DNA decay, the primordial organisms were endowed with universal endonucleases that dealt with both AP sites and U residues in a single step, as in the present-day NIR pathway (Figure 5). Nevertheless, today’s NIR-competent AP endonucleases remove AP sites more efficiently than modified bases. We propose that later in the course of evolution, due to a dramatic change in the spectra of DNA damage and for the sake of efficiency, uracil–DNA glycosylases substituted for endonucleases as the primary dU-recognizing enzymes, while the endonucleases evolved towards more stringent specificity for AP sites. Metal cofactors could have become the key limiting resource for endonucleases as life moved from the first poorly compartmentalized, easily perfused forms to tight self-contained cells; interestingly, EEP enzymes show excellent performance with many divalent metal ions (Mn^2+^, Co^2+^, Cd^2+^, Fe^2+^) but use catalytically less preferable Mg^2+^ due to easier availability [180]. With the separation of functions between DNA glycosylases and AP endonucleases, BER as we know it finally appeared.

Another testimony for the likely existence of universal endonucleases in LUCA is the remarkable specificity of today’s NIR for radiation-induced anoxic DNA base damage. The Earth’s magnetic field protects life from ionizing radiation coming in the form of solar wind and cosmic rays. Measurements from 3.3–4.2 Ga old minerals point to the weak magnetic field, about 1/10th of that at present [43,44]. Consequently, life on the ancient Earth was exposed to increased levels of ionizing radiation. As mentioned above, the spectra of radiation DNA damage drastically depend on the amount of oxygen dissolved in water. Under anoxic conditions, ionizing radiation preferentially generated DHT, DHU and αdN in DNA [49,51], which are preferred substrates for the NIR AP endonucleases such as archaeal Mth212, *E. coli* Nfo and human APE1 [26,163,181,182]. It should be noted that αdN in DNA cannot be recognized by DNA glycosylases, which require the presence of deoxynucleotides in β-anomeric configuration in order to flip them out from the DNA helix. Based on these observations, it is tempting to speculate that the AP endonuclease-initiated NIR pathway appeared very early in evolution as an adaptation to handle both spontaneous and induced DNA damage generated under anoxic conditions.

### 5.2. Specialized NIR: UVDE-Initiated Repair of UV Photoproducts and RNAseH2-Catalyzed Ribonucleotide Excision Repair

The oxygen-free atmosphere of ancient Earth offered much less protection from short-wavelength UVC radiation. Therefore, ancient forms of life that lived in surficial environments such as shallow ponds of condensed geothermal vapor were constantly exposed to UV light, which induces highly genotoxic CPD and 6-4PP adducts in cellular DNA. Although they are usually repaired by the DR and NER pathways, strikingly, these lesions are also substrates for endonucleases. Genetic and biochemical studies identified the UV damage endonuclease (UVDE) that recognizes and incises duplex DNA 5′ to CPD and 6-4PP [183,184]. Similar to NIR AP endonucleases, UVDE can cleave DNA duplexes containing AP sites, U, and DHU and clean single strand break termini from 3′-blocking groups to generate a 3′-OH end for DNA repair synthesis [185,186]. This UV damage endonuclease is not universally conserved and is found only in fungi (e.g., *Schizosaccharomyces pombe* and *Neurospora crassa*), Bacteria (though not in *E. coli*) and in a few archaeal lineages (Sulfolobaceae, Methanomicrobia and Halobacteria). The crystal structure of UVDE protein from the thermophilic bacterium *Thermus thermophilus* showed essential structural features of the TIM barrel fold very similar to that of Nfo [144,187], suggesting that UVDE probably evolved from the Nfo family AP endonucleases. Later, the crystal structure of archaeal UVDE from *Sulfolobus acidocaldarius* (SacUVDE) in a pre-catalytic complex with a DNA duplex containing a 6-4PP was resolved [188]. It revealed a novel dual dinucleotide flip mechanism for the recognition of bulky crosslinked dipyrimidines. SacUVDE flips the two purines opposite to the damaged pyrimidine bases into a dipurine-specific pocket, whereas the damaged pyrimidines are also flipped into another cleft [188]. Based on these observations, we propose that earlier in evolution the ancient life forms did not possess sophisticated repair systems such as NER machinery but rather employed either photolyases to remove UV-DNA adducts or NIR-like endonucleases to counteract spontaneous, radiation- and UV-induced damage.

Ribonucleoside monophosphates (rNMPs) can be erroneously incorporated in cellular DNA during DNA replication [189,190], and this problem was likely even more acute shortly after the transition from the RNA to DNA world. These misincorporated ribonucleotides are eliminated via RNase H2-catalyzed cleavage on the 5′ side of a ribonucleotide in DNA [191,192] to initiate a NIR-like pathway referred to as ribonucleotide excision repair (RER) [193]. Indeed, the mechanism of action of RNase H2 towards ribonucleotides in DNA is very similar to that of NIR AP endonucleases towards their substrates (Figure 1E,F). RNase H fold is shared by many endo- and exonucleases, including those participating in DNA repair, such as *E. coli* endonuclease V (see below), RuvC resolvase, UvrC NER endonuclease, and proofreading domains of A and B family DNA polymerases [194,195]. Based on this mechanistic resemblance, we propose that RNAse H2-mediated DNA incision is an ancient NIR pathway evolved to repair a distinct set of endogenous DNA damage that occurred in LUCA. Importantly, RNases H2 are present in all three domains of life and frequently found together with type 1 RNases H, which have the same fold and mechanism but cannot cleave single rNMPs embedded in DNA [195]. Noteworthy, many Archaea contain only type 2 RNases H [196]; furthermore, catalytic subunits of Archaeal and eukaryotic RNaseH2s have higher similarity to each other, both structurally and biochemically, compared with their bacterial counterparts [195].

## 6. Alternative Excision Repair Pathway: An Unusual Variation on the Endonuclease Theme

Endonuclease V (Nfi/EndoV) is another enzyme performing the cleavage of lesion-containing DNA. Nfi was originally isolated from *E. coli* in 1977 [197] and characterized as a Hx-specific enzyme in 1994 [198]. Nfi homologs are present in all domains of life and show activity towards DNA and RNA substrates. In contrast to Xth, Nfo and EndoQ endonucleases, the Nfi protein cleaves the second phosphodiester bond 3′ to a deaminated base. Further steps of the Nfi-mediated repair remain unclear, since the extension of the 3′-OH terminus will not lead to damage removal. The purified *E. coli* Nfi protein exhibits activities towards U, Hx, Xan, urea [198] and towards regular DNA containing mismatched bases, flap and pseudo-Y structures [199,200]. The Nfi-catalyzed activity towards Hx is ~20-fold higher than with any other DNA substrate [200]. The purified Nfi protein from hyperthermophilic bacteria *Thermotoga maritima* exhibits similar DNA substrate specificity to *E. coli* Nfi [201]. Interestingly, Nfi homologues are present in most Archaea except methanogenic species. EndoV from *P. furiosus* (PfuEndoV) is also similar to *E. coli* Nfi in its substrate preferences, but additionally recognizes and cleaves Hx-containing ssRNA, dsRNA and DNA/RNA hybrids with the same efficiency as the DNA substrates, suggesting a role in RNA editing [202]. PfuEndoV demonstrates much higher affinity and cleavage activity for Hx than other deaminated or mismatched bases in both ds- and ssDNA. *P. furiosus* also contains EndoQ (but no Hx-specific Family 6 UNG), which exhibits higher Hx cleavage catalytic efficiency and turnover rate and is apparently present in the cells at higher levels as compared to Nfi, suggesting that EndoQ-initiated NIR plays a major role in the removal of deaminated DNA bases in Archaea [203].

Despite the apparently higher specificity of Nfi towards Hx lesions, its function in *E. coli* remains controversial. An analysis of an *E. coli nfi* mutant showed an increased frequency of nitrous acid-induced mutations at A:T base pairs, suggesting a major role for Nfi in the in vivo Hx repair [204,205]. However, a later study found no increase in the level of Hx in the genomic DNA of *E. coli nfi* cells [206]. Indeed, the monofunctional alkylpurine DNA glycosylases AlkA in *E. coli* and MPG in human cells can efficiently remove Hx [207,208].

At present, the role of eukaryotic Nfi homologs in DNA repair is unclear, since these enzymes have no or very weak activity on DNA bearing Hx residues [209,210,211,212,213]. Instead, further studies revealed ribonuclease activity of human endonuclease V (ENDOV) on ssRNA and, with lower efficiency, dsRNA containing a single Hx residue [212,213]. In these substrates, ENDOV cleaves the second phosphodiester on the 3′ side of Hx, similarly to prokaryotic Nfi acting on DNA [212,213]. Importantly, *E. coli* Nfi is also able to cleave RNA substrates containing a Hx residue [213].

Although the Nfi-initiated “alternative excision pathway” remains on the list of minor repair schemes for specific lesions, Nfi after all may be not a true repair enzyme. To summarize the arguments against its role in DNA repair, (i) it is unusual among BER and NIR enzymes in the mechanism of DNA cleavage, which is not coupled to known repair cleansing activities and DNA strand displacement synthesis; (ii) it has non-specific DNA cleavage activity; (iii) it shows no DNA strand discrimination when incising duplex DNA with mismatches; and (iv) both prokaryotic and eukaryotic homologs have high affinity for RNA. The specificity of Nfi-like proteins for RNA loops and Hx-containing RNA may rather suggest their role in RNA processing and editing [213,214].

## 7. Oxygen Catastrophe: Back to BER

The known NIR-competent AP endonucleases can repair a wide variety of structurally distinct DNA lesions including oxidized pyrimidines, but neither highly mutagenic 8oxoG, the major oxidized base occurring in DNA under normoxic conditions, nor 8oxoA [123,163]. A huge increase in the oxygen concentration ~2.4–2.0 Ga is believed to have occurred due to the emergence of photosynthesis, leading to the so-called Oxygen Catastrophe, or Great Oxidation Event (GOE) [13,14]. The abundance of O_2_ in the atmosphere led to dramatic changes in the cellular metabolism and the appearance of oxidative phosphorylation that supplied sufficient amounts of energy for complex forms of life. However, leakage from the respiratory chain is an abundant ROS source, far exceeding the amounts produced by ionizing radiation. Moreover, oxygenated solutions produce other kinds of lesions than irradiation under anoxic conditions. Eventually, the evolution of life from oxygen-resistant to oxygen-dependent had to deal with new kinds of oxidized DNA bases, and also with complex DNA lesions such as bulky DNA adducts and inter-strand DNA crosslinks generated by the products of lipid peroxidation. GOE definitely post-dated the separation between Bacteria and Archaea. Regarding eukaryotes, it had long been held that their appearance broadly coincided with GOE; however, a recent integral analysis of genomic and fossil evidence put the emergence of eukaryotes after ~1.84 Ga [9]. In fact, due to multiple sinks, the accumulation of O_2_ in the Earth’s atmosphere cannot be described by a single oxidation event; at least two periods of O_2_ buildup are distinguished: a rise from nearly zero to ~0.02–0.04 atm pO_2_ ~2.4–2.0 Ga and a much more recent rise to the present-day ~0.2 atm pO_2_ starting ~0.7–0.8 Ga [13,14]. This complex profile might leave its footprint in the late diversification of DNA repair pathways such as the existence of multiple H2tH DNA glycosylases for oxidative lesions in multicellular eukaryotes.

Since LUCA existed long before GOE, solutions to combat the oxidative stress burden should have evolved in Bacteria and Archaea/Eukaryota independently, yet they had to be rooted in already existing pathways. An instructive census of archaeal BER/NIR glycosylases and endonucleases by Ishino and colleagues [166] shows that the most conserved and diverse DNA repair enzyme in Archaea is endonuclease III (Nth), an HhH, FeS-containing bifunctional DNA glycosylase that excises a wide variety of oxidized pyrimidines. Nth is also universally found in Bacteria, making it a good candidate for a possible ancestor of the HhH glycosylases (Figure 6). Nth can excise ionizing radiation-induced anoxic nucleobase lesions (DHT, DHU) and UV-induced pyrimidine hydrates [41,215,216], and might have been the primary enzymes for the repair of the latter damage type. Alternatively, or additionally, Nth-like HhH enzymes could be responsible for AP site repair using their AP lyase activity. AlkA, which is present in bacteria and many archaeal species, probably specialized in alkylation damage repair before the separation of these two domains. However, other oxidative damage HhH DNA glycosylases seem to have evolved after this split. MutY, which is uniquely suited for the post-replicative repair of 8oxoG, is found only in bacteria and eukaryotes. On the other hand, 8oxoG-specific DNA glycosylases OGG1 and OGG2 are restricted to Eukaryota and Archaea (with rare findings of OGG1 in Firmicutes, possibly due to lateral gene transfer), while bacteria employ Fpg, an H2tH 8oxoG DNA glycosylase.

So, if primordial DNA repair was probably performed by NIR-like endonucleases, why was it eventually supplanted by BER? One possible driving force could be the widened repertoire of lesions due to GOE. Since deaminated, alkylated and oxidative lesions are non-bulky and do not greatly distort DNA structure, there is no common mode of their recognition, as in NER. Ultimately, the specialization of repair enzymes towards their cognate lesions had to occur. A critical consideration in favor of specialized DNA glycosylases rather than 5′-endonucleases is that all glycosylases produce a common product whatever the initial lesion was. In effect, today’s BER has to deal with only three types of intermediates downstream of DNA glycosylases: AP sites (products of monofunctional glycosylases), 3′-terminal α,β-unsaturated aldehydes (products of β-elimination), and 3′-terminal phosphates (products of β,δ-elimination). On the contrary, specialized 5′-endonucleases would still leave the lesion in DNA, necessitating its removal by a 5′ → 3′ exonuclease or displacement during DNA repair synthesis. Many of today’s exonucleases and polymerases have problems encountering certain kinds of damage in the degraded or displaced strand, so in effect the specialized 5′-endonucleases would probably require specialized downstream repair enzymes to accommodate all kinds of DNA damage. Ultimately, the BER scheme with specialized DNA glycosylases followed by an AP endonuclease or a 3′-phosphodiesterase or a 3′-phosphatase turned out more streamlined and better suited for the removal of the most frequently occurring genome lesions.

## 8. Direct Repair as a Primordial Mechanism to Counteract Alkylation and UV Damage

Direct repair is a highly efficient and energy-saving mechanism that can remove a limited range of DNA lesions induced by alkylating agents and UV light. As compared to other more sophisticated DNA repair systems such as BER and NER, the DR pathway employs a single protein and does not require multi-protein machinery, or the excision of damaged nucleotides, or de novo DNA synthesis. Why does this simple scheme not apply to more types of lesions? We hypothesize that the reducing conditions on the primordial Earth limited induced DNA damage to only two types of bulky UV lesions and several DNA base modifications induced by alkylation agents and ionizing radiation. Therefore, it is reasonable to presume that at the beginning LUCA evolved the easiest mechanism to counteract just a few DNA base modifications. Moreover, since DR does not depend on template synthesis, the pathway may even pre-date the DNA world.

Alkylation damage of DNA induced by SAM radicals or nitrosamines at acidic pH was probably a major challenge to genome maintenance of ancient life. Metabolically activated nitrosamines react predominantly by an S_N_1 mechanism and preferentially generate 7meG, 3meA and *O*^6^meG residues. Due to their positive charge, 7meG and 3meA have weak glycosidic bonds and undergo accelerated depurination to AP sites, which can be efficiently repaired by AP endonucleases. On the contrary, *O*^6^meG, a highly mutagenic base, is quite stable even at elevated temperatures, and therefore presented a higher risk than other alkylated DNA bases. *O*^6^-alkylguanine DNA alkyltransferases (AGTs) that revert *O*^6^meG and *O*^4^meT back to guanine and thymine, respectively, belong to a family of highly conserved proteins that are widely represented in the three domains of life, but are apparently absent from plants. AGTs recognize the damaged nucleobase in DNA and irreversibly transfer the alkyl group to a Cys residue in their active site [217]. Since after the methyl transfer AGTs are inactivated, they are not true enzymes but rather suicidal proteins. Although the removal of a single lesion by AGTs requires one protein molecule at a time, this repair mechanism is highly conserved in evolution and present in higher eukaryotes, suggesting that AGTs are essential to preserve genome stability. Moreover, today’s FeS-dependent radical SAM enzymes include methyltransferases that use their Cys residues to transfer methyl groups [63], and they were also present in the LUCA genome [12], so it is quite possible that early AGTs were not suicidal.

*N*1-methyladenine (m1A) and *N*3-methylcytosine (m3C) are major toxic and mutagenic lesions induced by the S_N_2 type of alkylating agents in single-stranded DNA. In bacteria and eukaryotes, m1A and m3C are removed by oxidative demethylation catalyzed by α-ketoglutarate-dependent dioxygenases: AlkB and AlkB-like (ABH) proteins, respectively [218,219,220]. To repair these lesions, AlkB and ABH use Fe^2+^ to activate O_2_ and oxidize the methyl group, producing an unstable hydroxymethylated intermediate that spontaneously decays to yield formaldehyde and the original DNA base [221,222]. Another atom of the O_2_ molecule is consumed during the conversion of the α-ketoglutarate co-substrate to CO_2_ and succinate. The requirement of oxygen in AlkB-mediated direct repair suggests that oxidative DNA demethylation appeared later in evolution, after the Oxygen Catastrophe. Indeed, no AlkB homologues have been identified in Archaea, instead, as shown for *Archaeoglobus fulgidus*, m1A and m3C are removed through the BER pathway initiated by an AlkA-like DNA glycosylase [223]. This observation may suggest that the need for the removal of some alkylated bases from DNA could be one of the factors driving the evolution of early BER.

Photoreactivation is the visible light-dependent recovery from cytotoxic DNA damage caused by UVC and UVB. It was proposed that DNA photolyase-mediated photoreactivation was one of the first DNA repair pathways to evolve [1] and the most effective one in the repair of UV-induced CPD adducts [224]. DNA photolyases catalyze the conversion of UV pyrimidine dimers into a normal pair of pyrimidine bases driven by the absorption of photons with wavelengths 319–490 nm (reviewed in [225]). Photolyases belong to the large family of cryptochrome/photolyase proteins that are implicated in DNA repair, circadian photoreception, and transcriptional regulation. Members of the family are widely distributed in all three domains of life and can be divided in three classes according to their function: CPD-photolyases that repair CPD lesions, 6-4PP-photolyases that repair 6-4PP photoproducts, and cryptochromes. Cryptochromes do not participate in DNA repair, although they have probably evolved from photolyases, and instead employ their blue-light harvesting apparatus to mediate light-dependent growth in plants [226] and regulate the circadian clock in animals [227]. Photolyases are flavoproteins composed of two structural domains, an α/β HUP (HIGH-signature proteins, UspA, and PP-ATPase) domain and a helical domain [228,229]. They contain two light-absorbing chromophores: the catalytic two-electron-reduced deprotonated flavine adenine dinucleotide (FADH^−^) bound by the helical domain and a second chromophore bound by the HUP domain and acting as a light-harvesting antenna, which, in different species, can be either 8-hydroxy-7,8-didemethyl-5-deazariboflavin (8-HDF) [230,231] or methenyltetrahydrofolate (MTHF) [232,233]. The HUP domain has a very old ancestry pre-dating LUCA and rooted in a common nucleotide-binding precursor of photolyases, electron transfer flavoproteins, class I aminoacyl–tRNA synthetases, and nucleotidyl transferases [234]. Both flavins and MTHF were in the repertoire of cofactors likely used by LUCA [12], and it is possible that photolyases first appeared as RNA- or DNA-binding proteins that stabilized photoactive dimer-splitting molecules near DNA and later evolved into highly efficient two-chromophore enzymes. Within the cryptochrome/photolyase family, it has been suggested that its common ancestor might have been a flavoprotein involved in both UV damage repair and the light-dependent regulation of gene expression [4].

Unexpectedly, a recently discovered 6-4PP photolyase, PhrB from *Agrobacterium tumefaciens*, defines a new class of photolyases that lack 8-HDF and MTHF and seem to use an FeS cluster as a primary light-harvesting antenna [235]. This mechanistically connects photolyases with another group of UV damage DR proteins, so-called “spore product lyases”, which repair 5-thyminyl-5,6-dihydrothymine, a pyrimidine dimer that arises upon UV irradiation of dehydrated, A-form DNA [236]. These enzymes are typical FeS/radical SAM proteins, currently narrowly restricted to spore-forming bacteria, mostly Firmicutes [237]. While it is unclear whether spore product lyases are remnants of an early repair pathway or appeared later in evolution, their ancient protein fold is consistent with the idea that LUCA could already possess several ways to directly reverse UV damage.

## 9. Putative Origins of NER

Eisen and Hanawalt in 1999 and later on Goosen and Moolenaar in 2008 proposed that photolyases very likely appeared in evolution before the occurrence of more sophisticated NER pathways [1,2]. Although the NER machinery has an advantage over photolyases because of its broader repertoire and the ability for dark repair of UV lesions, bacterial taxa nearly universally contain both UvrABC proteins and DNA photolyases. This suggests that the main biological role of the NER pathway is not to repair CPDs, but rather to deal with other bulky helix-distorting DNA adducts induced by other factors than UV. Apparently, modern bacterial and eukaryotic NER pathways appeared only after the separation of bacteria and eukaryote domains [2].

NER by its nature is lesion-unspecific, repairing almost any lesion that strongly distorts the double helix. Strikingly, the NER machinery in eukaryotes has evolved completely independently from the bacterial and archaeal one. In eukaryotes, the central NER sensor, XPC/Rad4, descends from bacteriophage anti-restriction ArdC proteins fused with an archaeal-derived papain-like peptidase domain [238]. In bacteria, the same role is played by the UvrA_2_B complex, with both UvrA and UvrB belonging to a large and diverse family of P-loop NTPases that also includes MutS mismatch repair protein, RecA and many helicases [239]. In Archaea, only a few groups possess the UvrABC system, likely acquired from Bacteria via horizontal transfer, and some archaeal groups possess homologs of several eukaryotic NER proteins but no XPC, DDB2 or XPA [240]. This means that, even if the said homologs are involved in NER in Archaea, the organization of this machinery is very different from eukaryotic global genome NER (which requires XPC, DDB2, and XPA). The same could be said of transcription-coupled NER, since very few Archaea have CSB (although many have CSA). Nucleases involved in 5′- and 3′-incisions in NER can be reliably traced to other nuclease families. Eukaryotic Rad2/XPG/ERCC5, the 3′-incision nuclease, is closely related to FEN1 flap endonuclease, exonuclease I and 5′ → 3′ exonuclease domains of Family A DNA polymerases [241,242]. Both subunits of the 5′-nuclease, comprised of ERCC1 and ERCC4/XPF proteins, share a common fold with many proteins grouped together as PD-(D/E)XK family nucleases, including the MMR endonuclease MutH, specialized mismatch endonuclease Vsr, RecB and RecC components of RecBCD exonuclease, eukaryotic, archaeal and phage Holliday junction resolvases and a variety of restriction endonucleases [243,244]. Bacterial UvrC proteins use two domains to nick DNA: the 5′ incision domain is related to Nfi, RNases H, DNA polymerase A and B family 3′ → 5′ exonucleases, RuvC and phage transposases [245], whereas the 3′ incision GIY-YIG nuclease domain comes from a diverse group of proteins that comprises class I homing endonucleases, some II restriction enzymes and retrotransposon endonucleases [246]. It seems that eukaryotes and bacteria have independently adapted existing convenient nuclease modules to be guided by conformation-recognizing modules, also independently evolved in these two domains of life, and that in today’s Archaea we can see traces of these pre-existing nucleases as they were before the appearance of NER.

If NER in bacteria and eukaryotes evolved independently, with photolyases and UVDE already in place to deal with UV photoproducts, what could be the reason beyond NER appearance? NER should have played an important non-redundant function to take hold in two of three cellular life domains. In view of the NER ability to remove bulky lesions, a tenable explanation could be that at some point in life history, a new kind of bulky damage arose or became more widespread. Oxidative damage might not be the primary cause since 8,5′-cyclopurine lesions—the only class of oxidative lesions repaired predominantly by NER—are generated under anoxic rather than normoxic conditions. An interesting possibility is that NER could have emerged as a response to the ever more complex biogenic chemosphere. Today’s NER is critical to cope with DNA damage due to metabolically activated xenobiotics, such as benz[*a*]pyrene, aminofluorenes, aflatoxins, etc. Characteristically, even today specialized varieties of NER exist to counteract metabolically related DNA damage. For example, the genomes of Streptomycetes that produce doxorubicin and daunorubicin, two anthracycline intercalating antibiotics, encode the DrrC protein, a homolog of UvrA involved in the NER of lesions caused by these compounds [247]. In *E. coli*, Cho endonuclease is a homolog of the UvrC 3′-incision domain with a specificity for a subset of bulky adducts (cholesterol, menthol) and psoralen-induced interstrand cross-links [248,249]. The ability of NER to fight a wide range of lesions could become its critical advantage in a world full of more and more diverse genotoxic compounds.

## 10. Conclusions: “Nothing in Biology Makes Sense Except in the Light of Evolution”

Modern molecular biology took its origin after Watson and Crick’s discovery of the double helix structure of DNA, and since then made great achievements in the understanding of what life is. Molecular biology is inherently reductionist, trying to understand biological systems in terms of their components and ultimately reduce the complexity and diversity of living matter to universal physical and chemical principles. However, many questions in biology remain difficult to address solely in a reductionist way. For example, solving the atomic structure of a protein provides critical information on its functions, but offers little insight into its origins. We know that this protein fold can catalyze this reaction, but how was it selected from many other folds that could buttress the same arrangement of the reacting groups? On the other hand, evolutionary theory is inherently non-reductionist because of the randomness of variation and the principle of natural selection, which acts on whole organisms and populations. We can use sequences of genes and whole genomes to interpret large-scale patterns of evolution, perhaps up to the earliest steps of life, but cannot see with this information how the parts interact with each other in the cell and the body. Only by combining Darwinian principles of evolution with molecular biology can we arrive at a holistic approach to address the most interesting questions.

The DNA repair field had its start even before Watson and Crick, from the seminal works of Albert Kelner and Renato Dulbecco who independently discovered photoreactivation in the late 1940s. A wealth of molecular biology information accumulated since then allowed us to decipher mechanisms of DNA repair in the three domains of life and even in viruses. Now we have enough reductionist tiles to start the assembly of the evolutionary puzzle and understand the origins of repair pathways starting from (and maybe even before) LUCA. The analysis of the repair pathways in Archaea led us to propose that AP endonucleases and DNA damage reversal enzymes were the earliest DNA repair players that counteracted genotoxic effects of the spontaneous decay of DNA (deamination and base loss) and high-energy electromagnetic radiation (UV pyrimidine dimers and α-anomeric 2′-deoxynucleotides). Thus, AP endonucleases entered the repair field before monofunctional uracil–DNA glycosylases, and photolyases before sophisticated NER machinery. Yet, bifunctional DNA glycosylases exemplified by the Nth-like proteins probably appeared very early, together with AP endonucleases, but their activities were not directly coupled to NIR AP endonucleases and the modern BER pathway described in bacteria and eukaryotes appeared much later.

Primordial DNA excision repair mechanisms most likely appeared with the separation of functions between RNA and DNA, which in all probability pre-dated cellular forms of life. The primeval AP endonucleases and DNA glycosylases might have evolved with the appearance of DNA genomes, not only to protect them from spurious damage but also to eliminate non-standard bases, including U, Hx, pseudouridine, *N*^6^-methyladenine, 5meC and other base modifications, from the genome carrier.

We propose that the Great Oxidation Event played a crucial role in the origin of modern BER and NER pathways. The appearance of oxidative phosphorylation provided energy for the development of complex multicellular forms of life, but on the other side, extended the spectra of genotoxic DNA base damage. The increased structural variety of DNA damage resulted in the specialization of DNA repair enzymes towards their cognate lesions. This specialization resulted in the appearance of multiple DNA glycosylases which converted every distinct base modification into only three type of DNA repair intermediates, which are then taken care of by more specialized AP endonucleases/3′-repair phosphodiesterases, rather than by a single universal NIR AP endonuclease.

## Figures and Tables

**Figure 1 cells-10-01591-f001:**
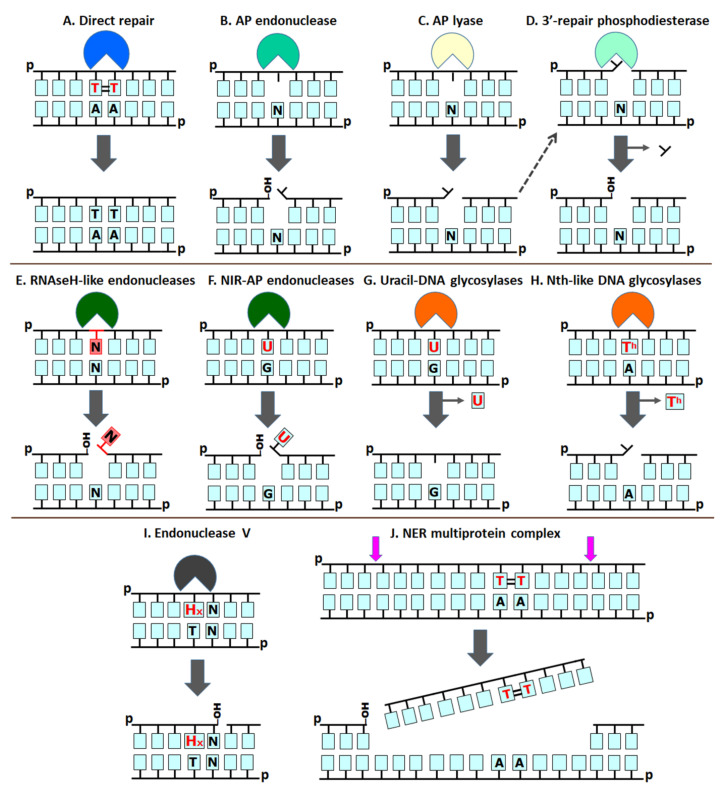
Principal DNA repair mechanisms employed by cellular organisms. (**A**) Direct repair of UV-induced pyrimidine dimer by a photolyase; (**B**) AP endonuclease-catalyzed cleavage of DNA at the 5’-side of an abasic site; (**C**) AP lyase cleaves DNA at the 3’-side of an abasic site; (**D**) 3’-repair phosphodiesterase and 3’-phosphatase remove a 3’-blocking moiety at DNA strand break generated by DNA glycosylases and free radicals; (**E**) RNAse H-like endoribonuclease cleaves a DNA-RNA duplex at the 5’-side of an RNA nucleotide; (**F**) NIR AP endonuclease cleaves DNA at the 5’-side of a 2’-deoxyuridine residue; (**G**) Uracil–DNA glycosylase removes uracil from DNA by hydrolysis of the N-glycosidic bond that links the base to the deoxyribosephosphate backbone, leaving an abasic site as the end product; (**H**) Nth-like bi-functional DNA glycosylases/AP lyases excise a pyrimidine hydrate in DNA and then cleave phosphodiester backbone at 3’-side of a remaining AP site; (**I**) Endonuclease V (Nfi) incises DNA at the second phosphodiester bond 3’ to a deaminated adenosine (hypoxanthine). (**J**) NER multi-protein complex removes a pyrimidine dimer in DNA by dual incision mechanism, in which the lesion is excised in the form of a short damaged nucleotides-containing single-stranded DNA oligonucleotide. N, deoxyribonucleotide; N with the red backbone, ribonucleotide; T^h^, thymine hydrate; U, uracil; Hx, hypoxanthine; T = T, UV induced pyrimidine dimers.

**Figure 2 cells-10-01591-f002:**
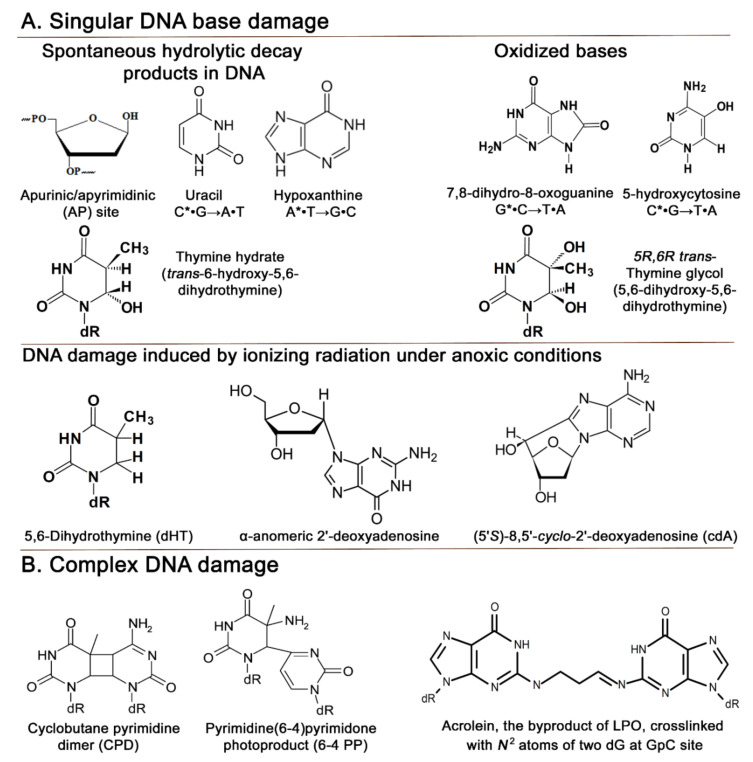
Chemical structures of spontaneous and induced DNA lesions. (**A**) Singular DNA base damage, including products of spontaneous DNA decay, oxidative DNA base damage and DNA base modifications generated by exposure to ionizing radiation under anoxic conditions. (**B**) Complex DNA damage, including bulky DNA adducts induced by UV light and inter-strand DNA crosslinks generated by reactive aldehydes.

**Figure 3 cells-10-01591-f003:**
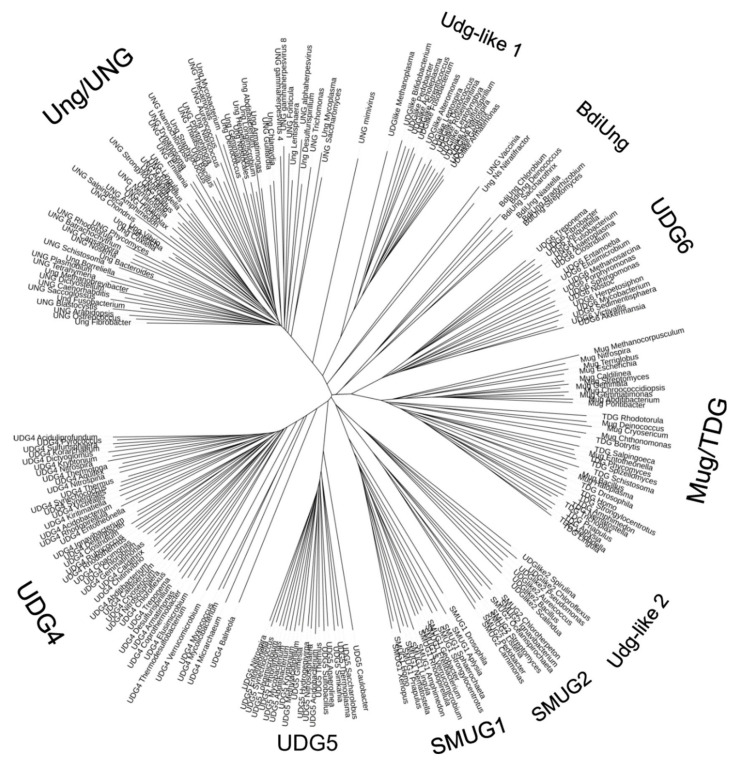
Unrooted tree of uracil–DNA glycosylases, limited to one per taxonomic phylum. The phylogenetic tree was produced by Clustal Omega [124] and visualized using iTOL [125].

**Figure 4 cells-10-01591-f004:**
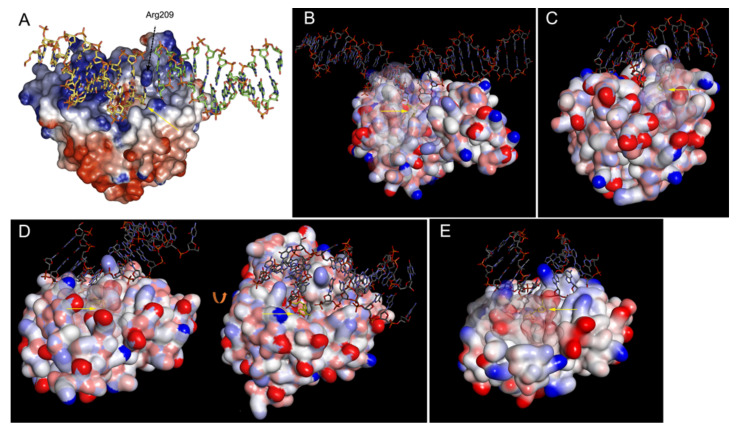
Nucleotide flipping out of DNA helix is the conserved DNA base damage recognition mechanism among NIR AP endonucleases. (**A**) Enzyme–substrate complex of the Mth212 D151N mutant obtained with dsDNA (exonuclease reaction substrate with a terminal dC in the active site) (PDB: 3GA6) [162]. (**B**) PfuEndoQ complex with a 27 mer duplex substrate with dU at the active site (PDB: 7K30) [167]. (**C**) Human APE1 bound to non-cleaved THF-DNA (PDB: 1DE8) [141]. (**D**) *E. coli* Nfo H69A mutant bound to a cleaved DNA duplex containing αdA (PDB: 4K1G) [168]. (**E**) *E. coli* Nfo Y72A mutant bound to THF-DNA (PDB: 2NQ9) [169]. Yellow arrows point to the nucleotides flipped into the active sites.

**Figure 5 cells-10-01591-f005:**
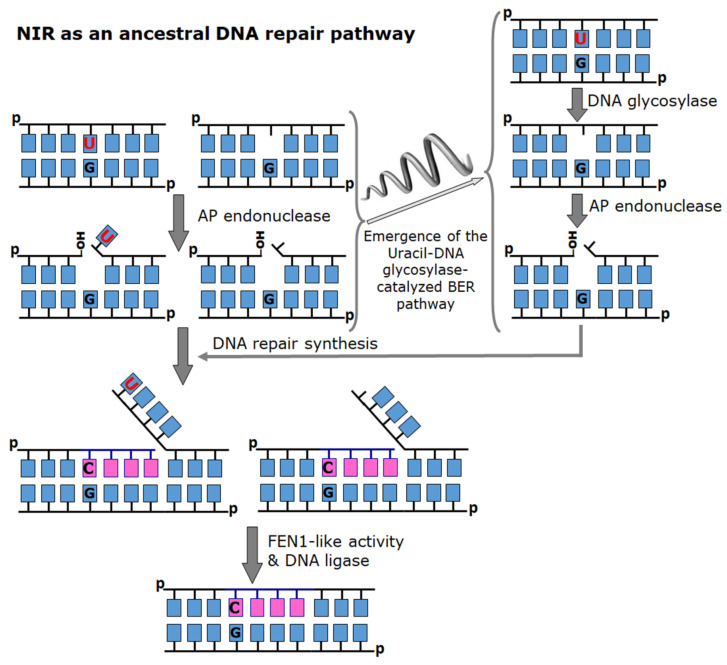
Putative evolutionary origin of the NIR and BER pathways for spontaneous DNA damage. In LUCA the products of spontaneous DNA decay were repaired by AP endonucleases in the NIR pathway in DNA glycosylase-independent manner. Mono-functional DNA glycosylases specific for uracil and hypoxanthine appeared later in evolution. Inclusion of additional base excision step into NIR pathway resulted in the appearance of modern BER pathway in three domains of life.

**Figure 6 cells-10-01591-f006:**
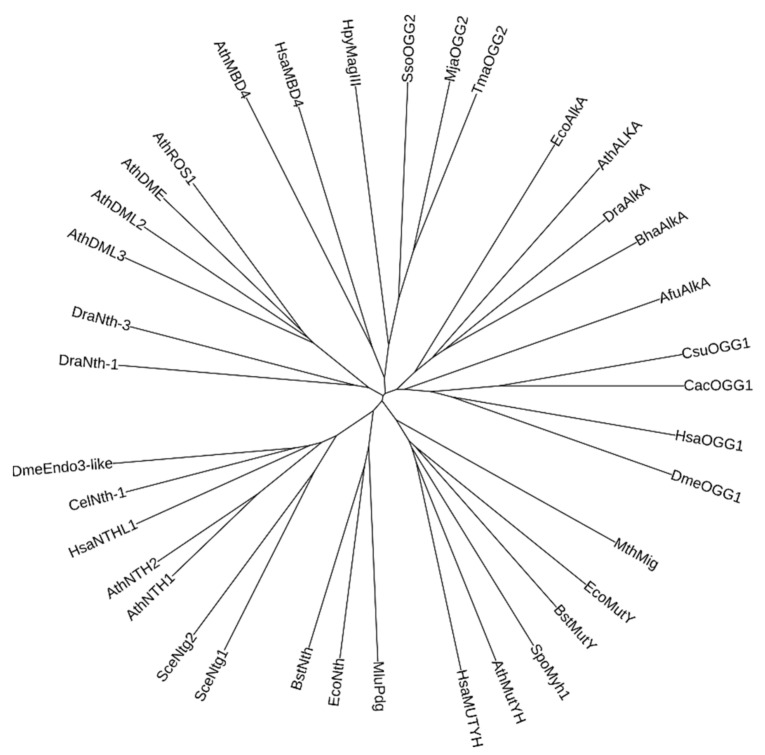
Unrooted tree of selected HhH DNA glycosylases. The species are: Afu, *Archaeoglobus fulgidus*; Ath, *Arabidopsis thaliana*; Bha, *Alkalihalobacillus* (formerly *Bacillus*) *halodurans*; Bst, *Geobacillus* (formerly *Bacillus*) *stearothermophilus*; Cac, *Clostridium acetobutylicum*; Cel, *Caernorhabditis elegans*; Csu, *Caldanaerobacter subterraneus* (formerly *Thermoanaerobacter tengcongensis*); Dme, *Drosophila melanogaster*; Dra, *Deinococcus radiodurans*; Eco, *Escherichia coli*; Hpy, *Helicobacter pylori*; Hsa, *Homo sapiens*; Mja, *Methanocaldococcus jannaschii*; Mlu, *Micrococcus luteus*; Mth, *Methanothermobacter thermautotrophicus* (formerly *Methanobacterium thermoformicicum*); Sce, *Saccaromyces cerevisiae*; Spo, *Schizosaccharomyces pombe*; Sso, *Saccharolobus (Sulfolobus) solfataricus*; Tma, *Thermotoga maritima*. The selected proteins are HhH DNA glycosylases characterized either biochemically or structurally. The phylogenetic tree was produced by Clustal Omega [124] and visualized using iTOL [125].

**Table 1 cells-10-01591-t001:** DNA repair protein-coding genes from reconstructed and minimal genomes.

Structural Superfamily	Process	*E. coli*	Human	Function *^a^*
LUCA genome reconstruction: Mirkin et al. [89] *^b^*				
*O*^6^-methylguanine-DNA–protein-cysteine methyltransferase	DR	Ada, Ogt, Atl1	MGMT	exocyclic *O*-alkyltransferases
NUDIX hydrolases	nucleotide pool sanitization	MutT	MTH1	oxidized NTP/dNTP hydrolysis
Uracil–DNA glycosylase	BER	Ung, Mug	UNG, TDG, SMUG1	uracil–DNA glycosylase
Helix–hairpin–helix	BER	Nth, MutY, AlkA	NTHL1, OGG1, MUTYH, MBD4	DNA glycosylases for various lesions
TIM barrel	BER	Nfo	–	AP endonuclease
5′ → 3′ exonucleases	various	PolA, Exo	FEN1	normal and damaged DNA degradation; flap endonucleases
Family A DNA polymerases; DnaQ-like 3′ → 5′ exonucleases	repair DNA synthesis, proofreading	PolA	POLN, POLQ	DNA polymerases; proofreading activity may or may not be present
DEAD-box helicases	NER, recombination repair	UvrD, RecB	PARPBP1	monomeric DNA helicases involved in recombination, replication, and repair
ATP-binding cassette transporters	MMR	MutS	MSH2–MSH6	mismatch detection
S5-like fold;histidine kinase-, DNA gyrase B-, and HSP90-like ATPase	MMR	MutL	MLH1, MLH3, PMS1, PMS2	mismatch signal transduction
P-loop NTPases	recombination repair; non-homologous end joining	SbcC	SMC3, RAD50	DSB recognition and/or processing
Calcineurin-like phosphoesterases	recombination repair; non-homologous end joining	SbcD	MRE11	DSB recognition and/or processing
Haloacid dehalogenases	recombination repair	RecJ	–	5′ → 3′ exonucleases
P-loop NTPases	recombination repair	RecA	RAD51, DMC1	strand exchange
LUCA genome reconstruction: Weiss et al. [12]				
PDDEXK nucleases	specialized mismatch repair	Vsr	–	T:G mismatch cleavage
PDDEXK nucleases	specialized mismatch repair	–	–	archaeal mismatch endonucleases (NucS/ EndoMS)
RNase H-like	AER	Nfi	ENDOV	deoxyinosine 3′-endonuclease
P-loop NTPases	recombination repair	RuvB	WRNIP1	hexameric DNA helicases driving branch migration during recombination
Family Y DNA polymerases	translesion synthesis	DinB, UmuC	POLH, POLI, POLK, REV1	DNA polymerases capable of translesion synthesis
DEAD-box helicases	various	RecQ	BLM, WRN	monomeric DNA helicases involved in recombination, replication, and repair
DEAD-box helicases	unknown, DNA damage-induced	Lhr	ASCC3	DNA damage-inducible DNA:RNA helicase
DEAD-box helicases	NER	RadD (YejH)	ERCC3	DNA helicase that functions in transcription and NER
DNA topoisomerases, subtype IA	possibly break repair and chromatin remodeling/maintenance	–	–	archaeal reverse gyrase; introduces positive supercoils into DNA
Radical_SAM	?	–	–	putative DNA modification/repair FeS-containing radical SAM protein
Minimal genome: Glass et al. [87] *^c^*				
Uracil–DNA glycosylase	BER	Ung	UNG	uracil–DNA glycosylase
TIM barrel	BER	Nfo	–	AP endonuclease
ATP-binding cassette transporters	NER	UvrA	–	NER lesion search
ATP-binding cassette transporters	NER	UvrB	–	NER lesion search
RNase H-like	NER	UvrC	–	NER nuclease
Minimal genome: Venetz et al. [88] *^d^*				
dUTPases	nucleotide pool sanitization	Dut	DUT	trimeric dUTP diphosphatase
Uracil–DNA glycosylase	BER	Ung	UNG	uracil–DNA glycosylase
Haloacid dehalogenases	recombination repair	RecJ	–	5′ → 3′ exonucleases

*^a^* Genes with an essential function in replication are not included except DNA polymerase I (PolA); *^b^* LUCA1.0 from [89]; *^c^* Minimal genome reconstruction based on combined data from [82,83,84,85,87]. The experimental gene essentiality data were from *Mycoplasma genitalium*, *M. pneumoniae* and *Bacillus subtilis* genomes; *^d^* Minimized *Caulobacter crescentus* genome.

## Data Availability

Not applicable.

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
