# Peer review of "Evolutionary Origins of DNA Repair Pathways: Role of Oxygen Catastrophe in the Emergence of DNA Glycosylases"

_cells, 2021, doi:10.3390/cells10071591_

Round 1

Reviewer 1 Report

The major thesis of this essay is that the rise of oxygen on the planet drove a diverse array of specialized glycosylases dedicated to the removal of oxidatively-induced base damage. The expert team of authors have done an outstanding job in weaving together a diverse set of threads into a coherent story in support of the major hypothesis. Overall, this provocative article is well written and provides an interesting perspective and will provide a great introduction to the hypothesis giving necessary background where appropriate. There are numerous places where, as I reader, I wanted more, but then this essay would have grown in length and overall the authors strike a good balance. The following are offered as points to consider:

  1. Many fine reviews have been published on the evolution of repair enzymes. For example the review by Eisen and Hanawalt, ref # 213. Perhaps a short section in the intro outlining several of these articles and the specific focus of this review would be helpful to the reader. Also for example the review by Krokan and Bjoras CSHP in Biol 2013 (which discussed convergence), and several papers by Susan Wallace and colleagues on the evolution of BER glycosylases would be helpful. With regard to photolyases Thomas Carrell and Muller have published a wonderful review (Curr Opin Struct Biol. 2009 Jun;19(3):277-85.)
  2. Line 70-73 – please give some real dates to the history of oxygen production here.
  3. 78 – In mentioning UV radiation here, it should be pointed out that prior to the rise of photosynthesis the earth has a reducing environment and without oxygen and an ozone layer UV-irradiation on the surface of the planet would have been highly deleterious to life. So while oxygen solved one problem it created another. It is this latter problem that this review is focused on.  This is described in more detail in section 2.2, but here a discussion here would be helpful.
  4. Figure 1 has been published in many forms in many reviews. I think a more classic view of Direct repair, BER, NER, NIR. Saying that APE, UDG and Nth pathways are unique in this manner is a bit misleading and should be under the general BER pathway. Furthermore Uvr ABC-like should be called NER.
  5. The survey of lesions with structures is good in Figure 2.
  6. Lines 187-188 – how and why did oxygen appear 2.4 Ga ago – what organisms were a live then? How do they compare to LUCA?
  7. Section 4, could be grounded in more unrooted trees like shown in Figure 5. Showing
  8. The structures shown in Figure 3 are well presented, but it should be pointed out that two important proteins in NER, UV-DDB and XPC also perform base flipping and the idea of base flipping as general recognition mechanism through hydrophobic stacking interactions was first described by Van Houten and Snowden in Bioessays in 1993 (PMID: 8466476) and later by Wei Yang (DNA Repair (Amst) . 2006 Jun 10;5(6):654-66.)
  9. Section 9, linkes 825-830. The speculation on why and when NER arose as compared to DNA photoylase is not warranted here, and the conclusion that NER evolved to deal with helix distorting lesions other than UV-induced photoproducts is wildly speculative.
  10. A unrooted tree on the evolution of NER proteins such as XPC/Rad4 versus UvrABC is warranted. Oddly some Archea have both.
  11. The logic at the bottom of page 9 regarding the evolution of NER is not well grounded in fact. What if the reverse was true? NER evolved to deal with UV-induced photoproducts and slowly over time revolved to have a much wider substrate specificity? What is the oldest organism known to have NER? What proteins are the most highly conserved? Why and how did TFIIH get coopted into serving both a transcription and DNA repair function?
  12. Starting with LUCA it would be good to show a evolutionary map perhaps in color for the four different pathways outlined at the beginning and then show how oxygen led to the expansion of specialized glycosylases in BER.
  13. One startling omission is the odd finding that C elegans has lost any capacity to deal with 8-oxoG, losing both OGG1 and MUTYH, as well as SMUG1? If oxygen was such a disruptive force in driving evolution why would nematodes lose these critical proteins? Or did other glycosylases expand their substrate repertoire, as was suggested in this paper by Sugiyam and colleagues: August 2009DNA Repair 8(7):844-51
  14. Perhaps two types of scientific inquires need to combine to truly understand the evolution of DNA repair enzymes – both sequence conservation, but also structural conservation and expansion as in the possible case for C elegans NTH?

Author Response

The major thesis of this essay is that the rise of oxygen on the planet drove a diverse array of specialized glycosylases dedicated to the removal of oxidatively-induced base damage. The expert team of authors have done an outstanding job in weaving together a diverse set of threads into a coherent story in support of the major hypothesis. Overall, this provocative article is well written and provides an interesting perspective and will provide a great introduction to the hypothesis giving necessary background where appropriate. There are numerous places where, as I reader, I wanted more, but then this essay would have grown in length and overall the authors strike a good balance. The following are offered as points to consider:

Answer to Reviewer 1. We would like to thank reviewer for interesting suggestions and arguments and nice appreciation of our work.

  1. Many fine reviews have been published on the evolution of repair enzymes. For example the review by Eisen and Hanawalt, ref # 213. Perhaps a short section in the intro outlining several of these articles and the specific focus of this review would be helpful to the reader. Also for example the review by Krokan and Bjoras CSHP in Biol 2013 (which discussed convergence), and several papers by Susan Wallace and colleagues on the evolution of BER glycosylases would be helpful. With regard to photolyases Thomas Carrell and Muller have published a wonderful review (Curr Opin Struct Biol. 2009 Jun;19(3):277-85.).

Answer 1. We agree with reviewer and we wrote a paragraph in Introduction section to discuss previous work of our colleagues. Following paragraph has been included in Introduction section to address the comment of the Reviewer 1, referencing the suggested articles: “Previously, several excellent bioinformatics studies discussed the origin of DNA repair. Perhaps the first phylogenomic analysis of DNA repair proteins, based on a limited set of the complete genome sequences, was performed by Eisen and Hanawalt who made predictions about the repair phenotypes and offered important insights into the evolution of whole pathways [1]. They suggested that photolyases, alkyltransferases, Xth/Nfo AP endonucleases and MutY/Nth DNA glycosylases are very ancient enzymes and were present in the common ancestor, whereas the NER pathway had different origins in bacteria and eukaryotes/archaea. A more recent phylogenetic study by Goosen and Moolenaar confirmed diverse origins of the NER pathway in bacteria and eukaryotes, and lack of the dedicated NER machinery in archaea [2]. Several recent review papers on the origin of photolyase/cryptochrome family describe evolution of these enzymes to perform diverse functions [3,4]. The convergent evolution of six structural superfamilies of DNA glycosylases is discussed in excellent reviews by Krokan’s and Wallace’s laboratories [5-7]”.

  1. Line 70-73 – please give some real dates to the history of oxygen production here.

Answer 2. We have introduced the dates as requested: “Abundant geological evidence agree on a dramatic rise in the atmospheric oxygen approximately ~2.4–2.0 Ga [13,14]”.

  1. 78 – In mentioning UV radiation here, it should be pointed out that prior to the rise of photosynthesis the earth has a reducing environment and without oxygen and an ozone layer UV-irradiation on the surface of the planet would have been highly deleterious to life. So while oxygen solved one problem it created another. It is this latter problem that this review is focused on.  This is described in more detail in section 2.2, but here a discussion here would be helpful.

Answer 3. We have introduced the following sentence on the role of oxygen in protection from UV light, as requested: “Later in history of Earth production of oxygen during photosynthesis resulted in the formation of the ozone layer, which efficiently shielded the life from lethal UV irradiation”.

  1. Figure 1 has been published in many forms in many reviews. I think a more classic view of Direct repair, BER, NER, NIR. Saying that APE, UDG and Nth pathways are unique in this manner is a bit misleading and should be under the general BER pathway. Furthermore Uvr ABC-like should be called NER.

Answer 4. We would like to thank reviewer for raising this important issue. Actually, Figure 1 depicts very schematic DNA repair reactions used in classic DNA repair pathway, rather than the complete chain of reactions in the BER, NER and NIR pathways. In our opinion, the classical representation of the repair pathways would be too voluminous and restrained, since ribonucleotide excision repair (RER), endonuclease V alternative excision repair (AER) and AP endonuclease-independent AP site repair (Fig. 1C and D) would be difficult to fit into general repair scheme of BER and NIR.

We do agree with reviewer on the replacement of “UvrABC-like endonuclease” by more general term “NER multiprotein complex”. To address this comment, we revised the sentences referring to Figure 1 in following manner: “As shown in Figure 1, cellular life evolved several distinct enzymatic mechanisms to counteract mutagenesis and DNA damage to their double-stranded genomes, such as photolyase-catalysed direct damage reversal (DR) (Figure 1A), removal of damaged residues via DNA excision (Figure 1B-J), specialized DNA polymerase-catalysed lesion tolerance/bypass, nucleotide pool sanitization by phosphorylases, and recombinational repair by RecA-like proteins”.

  1. The survey of lesions with structures is good in Figure 2.

Answer 5. We would like to thank reviewer for his/her appreciation.

  1. Lines 187-188 – how and why did oxygen appear 2.4 Ga ago – what organisms were a live then? How do they compare to LUCA?

Answer 6. This oxygenation event was due to the appearance of photosynthesis and the spread of photosynthesizing cyanobacteria. Obviously, LUCA lived much earlier, as discussed throughout the paper. We have added the mention of cyanobacteria to the last sentence of Section 2.2.

  1. Section 4, could be grounded in more unrooted trees like shown in Figure 5. Showing

Answer 7. We have added a figure (new Fig. 3, with other figures re-numbered accordingly) showing an unrooted tree of uracil–DNA glycosylases. Drawing more trees is possible but would not add much to the point of this section.

  1. The structures shown in Figure 3 are well presented, but it should be pointed out that two important proteins in NER, UV-DDB and XPC also perform base flipping and the idea of base flipping as general recognition mechanism through hydrophobic stacking interactions was first described by Van Houten and Snowden in Bioessays in 1993 (PMID: 8466476) and later by Wei Yang (DNA Repair (Amst) . 2006 Jun 10;5(6):654-66.)

Answer 8. We would like to thank the Reviewer for this interesting suggestion. We have added a parargaph about base flipping mechanisms by NER proteins and references: “The idea that repair proteins initially recognize the lesion via disruption of weakened base stacking and intercalation of aromatic amino acid side chains into the DNA duplex was proposed for the NER proteins [174,175]. Later, it has been demonstrated that the yeast XPC orthologue Rad4 bound to DNA containing a CPD adduct inserts a β-hairpin through the DNA duplex, which enables the two damaged base pairs to flip out of the double helix [176]. Noteworthy, similarly to UVDE, the everted non-damaged nucleotides opposite to CPD are specifically recognized by Rad4, whereas the nucleotides in the dimer become disordered. Furthermore, the crustal structure of the DDB1–DDB2 complex bound to DNA containing a 6-4PP adduct shows that a DDB2 hairpin inserts into the minor groove and extrudes the photodimer into a binding pocket [177]. This specific recognition mechanism enables DDB2 to capture lesions refractory to detection by other damage surveillance proteins. These data suggest that the lesions recognized by the NER proteins are characterized by thermodynamic instability, which in turn facilitates their flipping-out by the multi-protein complex machinery”.

  1. Section 9, linkes 825-830. The speculation on why and when NER arose as compared to DNA photoylase is not warranted here, and the conclusion that NER evolved to deal with helix distorting lesions other than UV-induced photoproducts is wildly speculative.

Answer 9. The idea of recent origin of modern NER pathways was proposed by Eisen and Hanawalt in 1999 (Eisen & Hanawalt, Mutat. Res. 435:171-213, 1999; see Figure 3 in the original paper) and later on has been supported by the phylogenetic study performed by Goosen and Moolenaar in 2008 (Goosen & Moolenaar, DNA Repair 7:353-79, 2008). Section 9 and other parts of our manuscript develop this interesting idea further. From the differences between bacterial and eukaryotic NER they seem to appear independently, and this raises the question why a second photoproduct repair pathway would be necessary while photolyases already existed. Thus, rather than being “wildly speculative”, our suggestion builds on the logic of the evolutionary process, as we explain in Section 9.

To address this comment we edited in the first paragraph of Section 9 as follows: “Eisen and Hanawalt in 1999 and later on Goosen and Moolenaar in 2008 proposed that photolyases very likely appeared in evolution before the occurrence of more sophisticated NER pathways [1,2]. Although the NER machinery has the advantage over photolyases because of its broader repertoire and the ability for dark repair of UV lesions, bacterial taxa nearly universally contain both UvrABC proteins and DNA photolyases. This suggests that the main biological role of the NER pathway is not to repair CPDs, but rather to deal with other bulky helix-distorting DNA adducts induced by other factors than UV. Apparently, modern bacterial and eukaryotic NER pathways appeared only after the separation of bacteria and eukaryotes domains [2]”.

Please also see the response to the Reviewer’s inquiry 11.

  1. A unrooted tree on the evolution of NER proteins such as XPC/Rad4 versus UvrABC is warranted. Oddly some Archea have both.

Answer 10. Since XPC and UvrABC are not trelated, there is not much sense in drawing and comparing these trees. As of July 16, 2021 a search in GenBank non-redundant protein sequence database revealed no archaeal homologs of XPC or XPA (at the quite permissive expectance level E = 10−9), while a few thousand homologs of UvrA, UvrB, UvrC, and UvrD were found in all major clades within Archaea.

  1. The logic at the bottom of page 9 regarding the evolution of NER is not well grounded in fact. What if the reverse was true? NER evolved to deal with UV-induced photoproducts and slowly over time revolved to have a much wider substrate specificity? What is the oldest organism known to have NER? What proteins are the most highly conserved? Why and how did TFIIH get coopted into serving both a transcription and DNA repair function?

Answer 11. We believe that here the Reviewer refers not to the bottom of page 9 (which is occupied by table 1) but to the end of Section 9. We of course agree that today NER is an important mechanism to cope with UV damage. However, evolution is economic and there are very few precedents when a system with redundant functions appeared and survived. Since NER in eukaryotes and bacteria apparently emerged with other ways of UV damage repair already in place, the simplest explanation is that its non-redundant function was to repair lesions of another kind. We definitely do not state that this scenario is true but at least it is a hypothesis worth of consideration. We have now outlined this rationale at the beginning of the last paragraph of Section 9:

If NER in bacteria and eukaryotes evolved independently, with photolyases and UVDE already in place to deal with UV photoproducts, what could be the reason beyond NER appearance? NER should have played an important non-redundant function to take hold in two of three cellular life domains. In view of the NER ability to remove bulky lesions, a tenable explanation could be that at some point in life history, a new kind of bulky damage arose or became more widespread”.

Regarding other question raised by the Reviewer in this inquiry, the answers would be either vague, or, at the present state of knowledge, even more speculative. The oldest organism known to have NER cannot be defined in terms narrower than Bacteria and Eukarya, which we amply discuss in this section. TFIIH, as a specific participant of the eukaryotic transcription complex, apparently got involved in NER after NER emergence in eukaryotes, as archaea have homologs of some TFIIH subunits but not TFIIH per se; perhaps its primary role was to help the RNA polymerase to traverse non-canonical secondary structures in large eukaryotic genes. We have added a comment regarding global genome NER and transcription-coupled NER to Section 9. Comparing conservation of proteins participating in NER in bacteria and eukaryotes would be possibly helpful to distinguish proteins that underwent a change of function during their evolution but, by itself, it is a topic for a research project rather than a review. We’re afraid that getting into discussion on the details of NER evolution would expand this review by a half or so; here, we would prefer to sacrifice depth for brevity and leave this part with some edits only to illustrate the idea that different repair pathways could supersede each other in dealing with different classes of lesions.

  1. Starting with LUCA it would be good to show a evolutionary map perhaps in color for the four different pathways outlined at the beginning and then show how oxygen led to the expansion of specialized glycosylases in BER.

Answer 12. We thank the Reviewer for this interesting suggestion. However, while the geological timeline of Earth oxygenation is well established, no studies so far addressed the chronology of appearance of DNA glycosylase superfamilies and splits between their subgroups. Again, such undertaking would be a theme for a research project rather than a review. So it is difficult to implement the proposed scheme here.

  1. One startling omission is the odd finding that C elegans has lost any capacity to deal with 8-oxoG, losing both OGG1 and MUTYH, as well as SMUG1? If oxygen was such a disruptive force in driving evolution why would nematodes lose these critical proteins? Or did other glycosylases expand their substrate repertoire, as was suggested in this paper by Sugiyam and colleagues: August 2009DNA Repair 8(7):844-51.
  2. Perhaps two types of scientific inquires need to combine to truly understand the evolution of DNA repair enzymes – both sequence conservation, but also structural conservation and expansion as in the possible case for C elegans NTH?

Answers 13 and 14. We note that almost any gene not belonging to housekeeping genes can be found missing in certain species of groups due to specific events in their evolutionary history that are rarely known. In the C. elegans example, we could not even say that nematodes as a whole lost a capacity to deal with 8-oxoG, because even the sequenced genomes of other nematodes belonging to the same order Rhabditida (Loa loa, Brugia malayi and Strongyloides ratti) contain OGG1. We agree with reviewer that it is surprising that C. elegans during their evolution lost essential elements of GO pathway, but it clearly occurred quite recently in evolutionary terms and is likely a quirk rather than a rule. Other well-known illustrations of strangely missing DNA repair pathways are the lack of uracil–DNA glycosylase in pupating insects (while conserved in other Hexapoda) and the lack of MutY-like DNA glycosylase and MutT-like hydrolase in S. cerevisiae (while conserved in many other ascomycetes including well-studied Schizosaccharomyces pombe). In the latter case, mismatch repair, Rad6/Rad18 and DNA polymerase eta systems efficiently compensate for the lack of MutY homologues in yeast. We can hypothesize that in C. elegans the NER machinery, MMR, NTH1 DNA glycosylase, and NDX-4 8-oxoguanosine phosphohydrolase could in combination provide a back-up for the loss of classical GO pathway. The question of variability of modern DNA repair pathways at the level of lower taxonomic ranks is very interesting in terms of their recent specific evolution, but its discussion is beyond the scope of our review that covers much earlier events in the life history. 

Reviewer 2 Report

Manuscript ID:  cells-1241250

Critique:

This is a well-written overview on evolutionary origins of DNA base excision repair pathways.  Authors point to the fact that a rise in the oxygen level in the atmosphere triggered a narrow specialization of AP endonucleases and DNA glycosylases to cope efficiently with arrays of oxidative base damage and even more complex DNA lesions. The review is unique, highly comprehensive, educational and enjoyable to read.

Author Response

Reviewer 2. This is a well-written overview on evolutionary origins of DNA base excision repair pathways.  Authors point to the fact that a rise in the oxygen level in the atmosphere triggered a narrow specialization of AP endonucleases and DNA glycosylases to cope efficiently with arrays of oxidative base damage and even more complex DNA lesions. The review is unique, highly comprehensive, educational and enjoyable to read.

Answer to Reviewer 2. We would like to thank reviewer for his/her nice comments and general appreciation of our work.